# Is Piezocision effective in accelerating orthodontic tooth movement: A systematic review and meta-analysis

**Samer Mheissen**[1]*, **Haris Khan**[2], **Shadi Samawi**[3]

**1** DDS, Syrian Board in Orthodontics, Former instructor in Orthodontic Department, Syrian Ministry of Health Private Practice, Damascus, Syrian Arab Republic, **2** BDS, FCPS, FFDRCSI, Associate Professor of orthodontics, CMH institute of dentistry Lahore, National University of Medical Sciences, Punjab, Pakistan, **3** BDS, MMedSci(Orth), MOrthRCSED, Private Practice, Argentina, Jordan

* Mheissen@yahoo.com

**Data Availability Statement:** All relevant data are within the manuscript and its Supporting Information files.

**Funding:** No funding was received for this review.

## Abstract

### Objective

This meta-analysis aimed at critically assessing currently available evidence regarding the overall effectiveness of Piezocision in accelerating orthodontic tooth movement, as well as the adverse effects of this intervention in orthodontic patients.

### Search methods

Electronic search of 6 databases and additional manual searches up to April 2019 without restrictions, also update the search was done by 20th November.

### Selection criteria

Randomized controlled trials (RCT) and controlled clinical trials (CCT) reporting piezocision-assisted orthodontics versus conventional orthodontics were included in the review.

### Data collection and analysis

The data are expressed by mean differences (MD), 95% confidence intervals, fixed-effect model or random-effect model in the meta-analysis in regard to statistical heterogeneity analyses (tau$^2$, and I$^2$). Included randomized studies were assessed for risk of bias using the new Cochrane Risk of Bias tool (ROB.2) and the non-randomized studies were assessed using (ROBINS I) tool. The studies were graded according to the GRADE approach.

### Results

Fourteen papers for 13 unique trials were included in this systematic review and eight studies were included in the meta-analysis. The meta-analysis showed that the mean difference of the canine retraction rate in the first and second month after piezocision was 0.66 mm/ month and 0.48mm/month, respectively. A total canine retraction rate in the first two months after piezocision was statistically significant (0.57 mm/month, p<0.00001), favoring the

**Competing interests:** The authors have declared that no competing interests exist.

piezocision group with a high heterogeneity between studies $I^2 = 69\%$. For the total treatment time outcome measure, there was a statistically significant difference in the overall treatment time (MD 101.64 Days, 95% CI, 59.24–144.06) favoring the piezocision group.

## Conclusions

Low quality evidence suggests that piezocision is an effective surgical procedure in accelerating the rate of canine retraction in the first two months and reducing the treatment duration. However, this effect appears to be clinically insignificant.

## Systematic review registration

CRD42019136303.

## Introduction

Malocclusion is a common problem of the maxillofacial region and has a global distribution. [1] It can negatively affect the quality of life of a person by compromising aesthetics and function. Patients with malocclusion can benefit from either removable or fixed orthodontic treatment, but treatment duration can range from months up to 2–3 years in case of comprehensive treatment[2] and is a matter of concern for the patient. Patient compliance to follow up orthodontic appointments decreases by 23% for every 6-month increase in treatment duration.[3] Also, longer treatment durations can increase the chances of iatrogenic damage like root resorption[4], white spot lesions [5], and periodontal problems.[6, 7]

Different non-surgical and surgical interventions have been used over the years to decrease the duration of orthodontic treatment. Non-surgical techniques include modification of biomechanics by customization in brackets and archwires, biological methods which include injection of different cell mediators, and device-assisted methods, which include vibrational stimulation, pulsed electromagnetic fields, low-level laser therapy, electric currents, and static magnetic field. [8] Surgical techniques include osteotomy or corticotomy procedures, interseptal alveolar surgery, micro-osteoperforations, corticision, discision, piezocision, and piezopuncture.

The surgical adjunct procedures work on the principle of regional accelerated phenomenon (RAP) first introduced by Frost.[9] RAP is evoked by noxious stimulus and is characterized by an increase in inflammatory mediators at the surgical site, which results in a decrease in bone density and an increase in bone resorption.[10] In RAP, increase osteoclastic activity was seen on the compression side, while increase staining of osteogenic markers was found on the tension side of orthodontic tooth movement in an animal study.[11] Among the surgical procedures used, piezocision is considered a safe adjunct[8, 12]procedure to rapid tooth movement showing more patient acceptability.

Vercellotti [13] in 2007 reported the first use of piezosurgery after conventional full thickness flap elevation for accelerating orthodontic tooth movement. Dibart [14] in 2009 recommended flapless corticotomies using a piezosurgical micro saw for making 3 mm deep incisions and coined the term 'Piezocision' for this procedure. In this technique, a BS1 cutting tip is used under copious irrigation to make an incision through the soft tissue and bone. The surgical incision is performed below the attached gingiva and is usually 5–10 mm long and 1 to 3 mm deep. A potential complication of this procedure involves root damage while performing the mucoperiosteal incision, as there is no direct visualization of the root position.

Radiographic metal guides placed on archwires have been advocated to avoid this complication.[15]

Many systematic reviews studied the effect of surgical and non-surgical adjunct procedures on the acceleration of orthodontic tooth movement, but none of them reported the pure quantitative effect of the piezocision.[12, 16–19]

## Objective

The current systematic review and meta-analysis aim to critically appraise the available evidence regarding the effectiveness of piezocision in accelerating canine retraction in the first two months after piezocision, alignment of teeth in crowded cases, en-masse retraction, treatment duration, as well as the adverse effects of this intervention in orthodontic patients.

## Material and methods

### Protocol and registration

This systematic review protocol was registered on PROSPERO (Registration Number: CRD42019136303). Review authors followed PRISMA statement[20] and the Cochrane Handbook for Systematic Reviews of Interventions [21] in reporting and conducting this review.

### Eligibility criteria

The reviewers have defined the eligibility criteria based upon the (PICOS) approach as follows:

**Participants.** Medically fit patients with any type of malocclusion, but without craniofacial anomalies or periodontal disease, from any age group in the permanent dentition, and requiring orthodontic treatment by fixed appliances.

**Intervention.** A piezoelectric device was used to perform corticisions for accelerating orthodontic tooth movement.

**Comparison.** Conventional orthodontics without any adjunct procedures for accelerating tooth movement.

**Outcome.** The primary outcomes: Canine retraction velocity measured in mm/month in the first two months, and the duration of the orthodontic treatment in relation to tooth alignment in crowded cases, en-masse retraction, and in maxillary incisors' retraction. The secondary outcomes: loss of anchorage, root resorption, gingival indices, and the patients' pain experience as assessed by the Visual Analogue Scale (VAS).

**Study design.** Randomized controlled trials (RCT) and controlled clinical trials (CCT) with a minimum of 10 participants.

**Exclusion criteria.** Studies were excluded for the following reasons:

- Animal and lab studies.

- Using burs for surgical cuts.

- Osteoperforation or any other adjunct surgical technique for acceleration.

- Decortication for rapid maxillary expansion.

- Patients taking medications that can affect tooth movement. e.g. prostaglandin Inhibitors and bisphosphonates.

- Involving participants who underwent orthognathic surgery by surgery-first approach.

- Individuals with craniofacial clefts or other syndromic conditions.

- Non-comparative studies from designs of cross-sectional, cohort studies, case series, or case reports.

## Information sources, search strategy and study selection

Electronic databases were searched up to 10<sup>th</sup> April 2019: Medline through PubMed, the Cochrane database, Cochrane Central Register of Controlled Trials (CENTRAL), and Scopus. [22] Also, for registered trials, we searched ClinicalTrial.gov and the International Clinical Trials Registry Platform (ICTRP). For further records, references of the included studies were checked. The principal author (S.M) has developed the search strategy for Medline using the PubReMiner tool [23, 24], and the search strategy for each database was based upon Medline search strategy with respect to differences in controlled vocabularies among databases. (Table 1 and S1 Table)

The search was not restricted to language, publication year, or initial malocclusion. Also, the updated PubMed search was done up to the 20<sup>th</sup> November 2019.

Two reviewers (S.M and H.K) performed the search and assessed the titles and the abstract for inclusion independently and in duplicate. Again, they assessed the full text for eligibility criteria independently with an excellent agreement of 85.7%, according to Kappa statistical analysis. The disagreement was resolved with discussion and consultation with the third author (S.S).

## Data item and collection

Two reviewers (S.M and H.K) extracted the data from the included studies independently and in duplicate. The disagreement was resolved by a joint discussion with the third author (S.S). Predesigned data extraction forms were used to describe the included study information.

## Risk of bias in individual studies

The same two reviewers (S.M and H.K) assessed the risk of bias in duplicate for the included studies and independently, using ROB.2 Cochrane risk of bias tool for the randomized controlled trials and ROBINS-I tool for non-randomized controlled trials.[25, 26] Any disagreement was resolved by joint discussion with the third author (S.S).

## Summary measures and approach to synthesis

The researchers pooled data in this meta-analysis from studies that were similar in participants, intervention, and outcomes. For the quantitative synthesis, the treatment effect with the

**Table 1. Search strategy.**

| Database | Number of records |
|---|---|
| PubMed | 804 |
| Cochrane | 88 |
| Scopus | 984 |
| Trial.gov | 108 |
| ICTRP | 2 |
| Updated Search | 10 |
| Other sources | 1 |
| Sum of records | 1997 |

results for continuous outcome was expressed as mean difference (MD) with 95% confidence interval (CI).[21] The impact of the heterogeneity between studies was detected by $Tau^2$ and $I^2$ statistics in RevMan 5.3 software. The researchers used a fixed- effect model for the meta-analysis when $I^2$ was under 50% and a random-effect model when $I^2$ was above 50% and there was a substantial heterogeneity between studies.

## Risk of bias across studies and additional analyses

If the number of studies was sufficient, we planned analyses for small-study effects. Wherever possible, subgroup analyses were based on characteristics of intervention and measurement scales. The GRADE approach was followed to assess the quality of evidence.[27, 28]

# Results

## Study selection and characteristics

The electronic searches resulted in 1986 records, plus one record identified through other sources and 10 records from updated PubMed search (Fig 1, Table 1). After removing duplicates, 1728 articles remained. When reviewing the remaining 1728 titles and abstracts, 30 references were identified, and 1698 records were excluded on the basis of their title and abstract. However, six studies were registered trials and not published. The authors of these studies were contacted but most of them failed to respond while others refused to share their data as their registered trials were still in progress. After the critical full text reading, 16 studies were omitted because they used burs or other types of corticision or because of the study design (S2 Table), One of those was a Chinese study and was excluded because of poor methodology, and one other record was a thesis for a published included paper [29]. Finally, 14 papers for 13

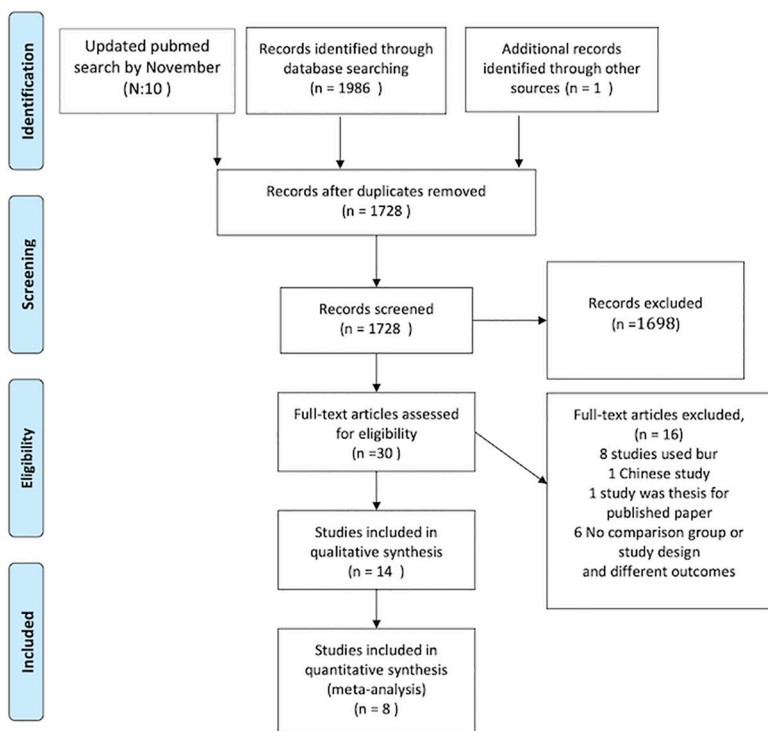

**Fig 1. Flow diagram for included articles.**

unique trials were included in this systematic review. For the quantitative analysis, eight out of the final 14 studies were pooled in the meta-analysis. Of those eight studies, one trial [30] did not use clear baseline parameters, and the measurements of the outcome were not suitable. One study[15] had a discrepancy in the data, while two papers reported the same trial, and one study[31] used different statistical distribution.

## Risk of bias within studies

The summary findings are presented in Fig 2, Table 2, and S3 Table.

Most of the RCTs were classified as having a high risk of bias. Only one study[32] was judged as having a low risk of bias, and three studies[33–35] were judged in the "some concerns" category. The two CCTs were having a serious risk of bias because of a lack of information regarding missing data.

Some studies [36, 37] failed to treat the missing data, and other studies [15, 31, 33, 35, 38–40]had some concerns in treating missing data. The randomization process was in some concern in five studies[29, 31, 35–37] and at low risk of bias for other studies[15, 32, 33, 38–41].

Measurement of the outcome was classified as having a high risk of bias in five studies, [15, 31, 37, 38, 41]because of the lack of blinding of the assessors and the method of measurement.

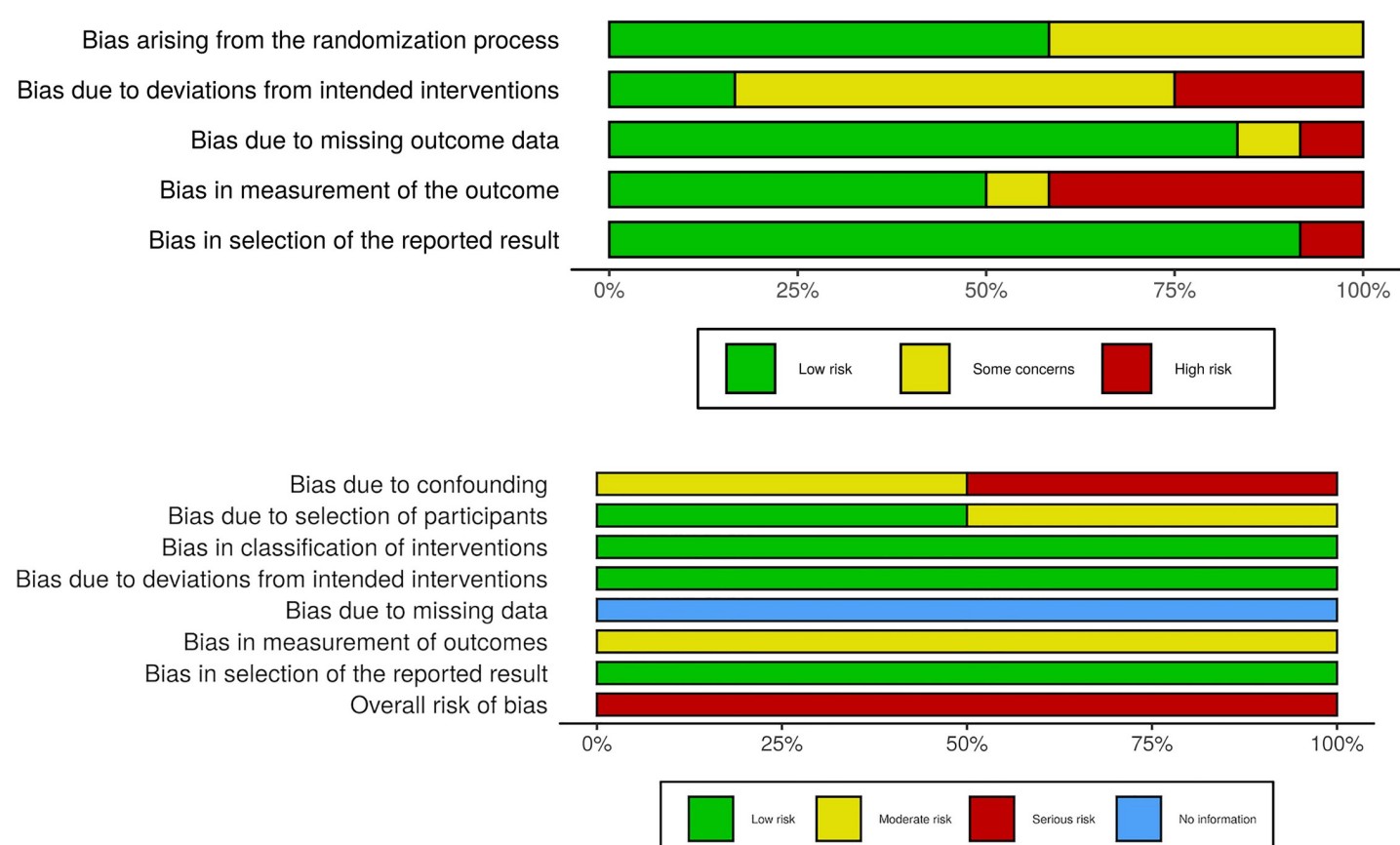

**Fig 2. A**. Summary of Risk of bias assessment for randomized controlled trials using ROB.2 tool. **B**. Summary of Risk of Bias assessment for non-randomized controlled trials using ROBINS-I tool.

**Table 2. A; Summary of risk of bias assessment for randomized controlled trials.** B; Summary of risk of bias assessment for non-randomized controlled trials.

**A**

| Study | Randomization process | Deviations from intended | Missing outcome data | Measurement of the outcome | Selection of the reported result | Overall Bias | | |
|---|---|---|---|---|---|---|---|---|
| Uribe | Some concerns | High risk | High risk | Low risk | Low risk | High risk | | |
| Tuncer | Low risk | Some Concerns | Low risk | High risk | Low risk | High risk | | |
| Abbas | Some concerns | High risk | Low risk | Some Concerns | Low risk | High risk | | |
| Aksakali | Some concerns | Some Concerns | Low risk | Low risk | Low risk | Some Concerns | | |
| Alfawal | Low risk | Some Concerns | Low risk | Low risk | Low risk | Some concerns | | |
| Charavet 2016 | Some Concerns | Some Concerns | Low risk | High risk | Low risk | High risk | | |
| Charavet 2019 | Low risk | Low risk | Low risk | Low risk | Low risk | Low risk | | |
| Charavet (PROMs) 2019 | Low risk | Low risk | Low risk | High risk | Low risk | High risk | | |
| Gibreal 2018 | Low risk | Some Concerns | Low risk | Low risk | High risk | High risk | | |
| Gibreal 2019 | Low risk | Some Concerns | Low risk | High risk | Low risk | High risk | | |
| Al-Imam 2019 | Low risk | Some Concerns | Low risk | Low risk | Low risk | Some Concerns | | |
| Raj 2020 | Some Concerns | High risk | Some Concerns | High risk | Low risk | High risk | | |

**B**

| Study | Bias due to confounding | Bias in selection of participants into the study | Bias in classification of interventions | Bias due to deviations from intended interventions | Bias due to missing data | Bias in measurement of outcomes | Bias in selection of the reported result | Overall |
|---|---|---|---|---|---|---|---|---|
| Wu | Moderate | Low | Low | Low | NI | Moderate | Low | Serious |
| Yavuz | Serious | Moderate | Low | Low | NI | Moderate | Low | Serious |

## Results of individual studies, meta-analysis, and additional analysis

Based on the outcome measures in the included studies, we pooled the data into two categories; Canine Retraction and Treatment Time Duration.

### Canine retraction

Four split-mouth randomized controlled trials [33, 35–37] studied the acceleration of canine retraction. Three trials [33, 35, 36] used 0.022× 0.028- inch slot brackets. Also, most of them [35–37]used 0.016×0.022-inch SS archwires for retraction, except Alfawal et al.[33] who used 0.019× 0.025-inch SS archwire for canine retraction. (Tables 3 and 4)

The meta-analysis showed (Fig 3) that the mean difference of the canine retraction rate in the first two months after piezocision was statistically significant; 0.57 mm/month (95% CI, 0.42–0.71, p<0.00001) and favored the piezocision group with a high heterogeneity between studies $I^2 = 69\%$. Also, there was no statistically significant difference in the movement rate between the first and the second month after the piezocision surgery (Fig 3). Molar anchorage loss was more in the control group (MD 0.53mm, 95% CI, 0.05–1.01 mm, p = 0.03, $I^2 = 72\%$) with a high heterogeneity. (Fig 4)

### Treatment duration

**En-masse retraction.** Two trials[38, 42]studied the acceleration of en masse retraction. However, Tuncer et al.[38] used 0.016×0.022-inch SS archwire for retraction with bi-

**Table 3. Results summary table showing the quality of the evidence according to GRADE approach.**

Piezocision compared to Control in Orthodontic

Patient or population: Orthodontic

Setting:

Intervention: Piezocision

Comparison: Control

| Outcomes | of participants (studies) Follow-up | Certainty of the evidence (GRADE) | Relative effect (95% CI) | Anticipated absolute effects | |
|---|---|---|---|---|---|
| | | | | Risk with Control | Risk difference with Piezocision |
| Canine Retraction Rate assessed with: MM/Month | 81 (4 RCTs) | ⊕⊕◯◯ LOW [a,b,c] | - | | MD **0.57 mm/month more** (0.42 more to 0.71 more) |
| En masse retraction assessed with: Months | 55 (2 RCTs) | ⊕⊕◯◯ LOW [a,b,c] | - | | MD **4.3 month lower** (4.23 lower to 12.48 lower) |
| Total Treatment Time assessed with: Days | 45 (2 RCTs) | ⊕⊕◯◯ LOW [a,b,c] | - | | MD **101.64 Day lower** (59.23 lower to 144.06 lower) |

*The risk in the intervention group (and its 95% confidence interval) is based on the assumed risk in the comparison group and the **relative effect** of the intervention (and its 95% CI).

CI: Confidence interval; MD: Mean difference

GRADE Working Group grades of evidence High certainty: We are very confident that the true effect lies close to that of the estimate of the effect

Moderate certainty: We are moderately confident in the effect estimate: The true effect is likely to be close to the estimate of the effect, but there is a possibility that it is substantially different

Low certainty: Our confidence in the effect estimate is limited: The true effect may be substantially different from the estimate of the effect

Very low certainty: We have very little confidence in the effect estimate: The true effect is likely to be substantially different from the estimate of effect

dimensional brackets, while Wu et al. [42] used 0.022× 0.028- inch slot brackets with 0.019× 0.025-inch SS archwire during en-masse retraction. (Tables 3 and 4).

The pooled estimate for en-masse retraction resulted in 4.30 months (CI 95%, 4.23–12.48, P = 0.32), but it was not statistically significant with a very high heterogeneity between the two studies $I^2$ = 96% (Fig 5).

**Decrowding.** Two RCTs[31, 32] and two CTs[42, 43] compared the overall treatment duration for leveling and alignment between the piezocision group and control group using self-ligating brackets. Only Uribe et al.[29] assessed the alignment time.

We pooled two homogenous studies[32, 43] in the meta-analysis. There was a statistically significant difference in the overall treatment time (MD 101.64 Days, 95% CI, 59.24–144.06) favoring the piezocision group with a statistical homogeneity $I^2$ = 0 (Fig 6).

**Incisors' retraction.** Only one study[40] reported the acceleration of incisors' retraction with assistance of piezocision cuts from the palatal and labial aspects. The authors concluded that the piezocision accelerated the incisors' retraction and decreased the time of retraction by 27% when compared to incisors' retraction without piezocision procedure.

## Adverse effects

**Root resorption.** Abbas et al.[36] concluded that root resorption was higher in the control group. Two studies[31, 32] reported that there was no increase in root resorption, and there was no significant increase in fenestration or dehiscence of the roots. Also, Raj et al.[37] reported statically significant root resorption in canines in the piezocision and control sides without differences between them in the six months follow up.

**Periodontal score.** Two studies [35, 43] reported that there was no statistical difference between the two groups regarding gingival indices and mobility scores. Aksakalli et al.[35]

**Table 4. Included studies' extraction table.**

| Study | Design, settings | Participants | Interventions / Comparison | Outcomes | Follow-up |
|---|---|---|---|---|---|
| Abbas, N. 2016 | S-M, RCT Faculty of dentistry Egypt | N: 10 patients Class II Div1, mild or no crowding; age: 15–25 | Roth 0.022 AW 0.016 × 0.022-in SS. CCS 150 g **Intervention**: RF, | Canine movement rate Anchorage loss Periodontal health canine root resorption | 3 months after the start of the canine retraction 2 weeks interval |
| Aksakalli 2016 | S-M, RCT University Istanbul, Turkey | Class II malocclusion N: 10 (6 F:4M) Mean age: 16.3 ± 2.4 y | Roth .022- AW 0.016 × 0.022-inch SS EC 150 g **Intervention:** 2 Cuts, PD 3 mm Flapless | canine and molar movement rate Mobility scores and gingival indices for the canines transverse changes | 2-week intervals |
| Wu 2015 | A pilot clinical study Peking University China | Class III N: 24; age: 18–30 Upper mild crowding | Edgewise 0.022 AW 0.019× 0.025-inch SS. EMR **Intervention**; FR. PD: cortical plate thickness BG | Leveling and alignment time EMR rates Total orthodontic treatment time | 4 weeks |
| Tunçer 2017 | RCT Başkent University Turkey | Class I or Class II N: 15 per group, (13 F:2 M); age: PG 17.7y, CG 17.0y | AW 0.016 × 0.022, Bidimensional Brackets CCS 250g Flapless PD 3 mm | EMR rates of maxillary anterior teeth | day 15, 30, 60, 90 and 120. 9.3 months of follow-up |
| Alfawal/ 2018 | S-M, RCT University, Syria | Class II div I N: 17, (11 F, 7 M) age: 18.70 ± 3.6 | MBT 0.022. AW sequences: 0,014. Or 0.016. 0,016 × 0,022, 0,017 × 0,025 in. NiTi, 0,019 × 0,025 in. ss. **Intervention**: PD 3mm CCS 150-g force | The rate of canine movement, molar anchorage loss, canines' rotation and the duration of canine retraction, | 2 weeks interval 1, 2, 3and 4-months measurement taken |
| Uribe 2017 | RCT Uni, US | LLI > 5 mm of mandibular anterior crowding Age: CG 29.4 y (6 M, 7 F) PG: 30 y (6 M, 10 F) LII >5 CG 8.32 (2.29) PG 6.73(1.99) | SL Carriere brackets V. Aw sequence 0.014, 0.014 × 0.025-inch CuNiTi **Intervention:** 3 PC, PD 1mm Flapless | Orthodontic outcomes Little's irregularity index was used to measure the amount of crowding on the dental models at every appointment. | Follow up monthly (every 4–5 weeks) Final irregularity index was ≤2 mm |
| Charavet 2019 | RCT Uni, Belgium. | N: 24 (15 F:9M) LII< 6 Mean age: PG 29 ± 8 CG 27 ± 7 | CAD/CAM SL appliances AW sequence 0.014-in, 0.018-in, 0.014 × 0.025-in, 0.018 × 0.025-in CuNITI, 0.019 × 0.025-in SS. **Intervention:** FR, PD 3mm, PL 5mm | Overall treatment duration periodontal parameters, gingival scars Radiographic root resorptions, dehiscence and fenestration scores using | Every 2 weeks and archwires were changed only when full bracket engagement was achieved. |
| Charavet 2019 PROMs | RCT Uni, Belgium. | N: 24 (15 F:9M) LII< 6 Mean age: PG 29 ± 8 CG 27 ± 7 | CAD/CAM SL appliances AW sequence 0.014-in, 0.018-in, 0.014 × 0.025-in, 0.018 × 0.025-in CuNITI, 0.019 × 0.025-in SS. **Intervention:** FR, PD 3mm | patient-centered outcomes Level of apprehension Pain level Paracetamol consumption—Patient satisfaction | |
| Charavet 2016 | RCT Uni, Belgium | N: 24 (9 M, 15 F) age: CG 27 ± 7 PG: 34 ± 8 II <6 Maxilla PG: −2.8 ± 1.2 CG: −2.3 ± 1.5 Mandible PG: −3.4 ± 1.4 CG: −2.6 ± 1.8 | DSL (Ormco) AW sequence: 0.014-in, 0.018-in., 0.014 × 0.025–in, 0.018 × 0.025–in. Cu NiTi. 0.019 × 0.025–in. ss. **Intervention:** FR PD 3mm PL 5mm | The total Treatment time Periodontal Parameters: Radiographic Root resorptions, dehiscence and fenestration score. | Every 2 weeks. Changing archwires when they were no more active. |
| Gibreal 2018 | RCT Uni, Syria | N: 36, SC age: 16–27 (20.32) CG: 7 M 10 F PG: 8 M 9 F LII = 10 mm | MBT 0.022. AW sequence: 0.014-inch, 0.016-inch, 0.016 × 0.022-inch, 0.017X 0.025-inch NiTi, 0.019 × 0.025-inch ss. **Intervention:** Flapless PD 3mm PL 5-8mm | The overall alignment time of the lower arch | Every two weeks. Changing archwires when they were passive. |
| Yavuz 2018 | CT Uni, Turkey | Class I, M-SC, NE N: 35 F CG; n = 14; age 13–19 y PG; n = 9; aged 13–18 y LII 10.48 (6.16) Mndible 6.47 (3.87) | Roth SLB 0.022. AW sequence: 0.014-in., 0.016-in., 0.018-in., 0.016 × 0.022-in., 0.017 × 0.025-in. NiTi 0.019 × 0.025-in.ss. **Intervention:** Flapless PD 3mm PL7 mm | LII scores Periodontal measurements VAS The total orthodontic treatment durations | 2–3 weeks intervals. |
| Raj 2020 | S-M, RCT Uni, Cuttack | Class II N: 20, age 23.18 ± 1.41 y | prescription: NI **Intervention:** Flapless PD 3-mm CCS 150 mg on 0.016*0.022 SS wire. | The canine and molar movement rate Periodontal indexes ABL Root resorption | Follow up period of 7 months. The patients were seen after surgery by at 1, 3, and 6 months for periodontal assessment and every two weeks for canine retraction |

*(Continued)*

**Table 4.** (Continued)

| Study | Design, settings | Participants | Interventions / Comparison | Outcomes | Follow-up |
|---|---|---|---|---|---|
| Gibreal 2019 | RCT Uni, Syria | N:34, SC, EX; age: 17–24 (21.03) CG: 7 M 9 F PG: 6 M 10 F | MBT 0.022-inch. AW sequence: 0.014-inch, 0.016-inch, 0.016 × 0.022-inch, 0.017X 0.025-inch NiTi, 0.019 × 0.025-inch ss. **Intervention:** Flapless PD 3-mm PL 5-8mm | levels of pain, discomfort and patients' satisfaction using visual analog scales | Patient called at every two weeks. Changing arch wires when they were passive. |
| Al-Imam 2019 | RCT Uni, Syria | Class II N: 42, age 15–26 y. | MBT 0.022-inch. AW sequence: 0.014-inch, 0.016-inch, 0.016 × 0.022-inch, 0.017X 0.025-inch NiTi, 0.019 × 0.025-inch ss. NiTi CCS; 150 g **Intervention:** Flapless PD 3-mm from the palatal and buccal aspects | Rate of incisor retraction and time required for retraction. Molar anchorage loss | The patients were seen at 3 weeks intervals. |

S-M: split mouth design, RCT: randomized controlled trial, PG: piezocision group, CG: control group, Uni: University settings, F: female, M: male, N: number, SC: severe crowding FR: Flap raised, PD: Piezocision depth, PL: Piezocision length, CCS: closed coil spring, EC: elastic chain, AW: Archwire ABL: alveolar bone level, SL: self-ligating, DSL: Damon self-ligating brackets. EMR: En-masse retraction

indicated a slight increase in mobility scores in both groups. However, there was no increase in the overall recession score in the groups, and more than 50% of piezocision patients had noticeable scars. [31, 32, 41] Also, Raj et al.[37] concluded that the probing depth and the relative attachment level (RAL) increased gradually in both sides without statistical differences between the piezocision and control side. Moreover, Al-Imam et al.[34]reported that they excluded one patient who had acute postsurgical inflammation between the upper central incisors from the palatal aspect.

**Pain in patients reported outcome measures.** The visual analogue scale (0–10) after piezocision surgery for pain level was (6.0 ± 1.9) and (6.8 ± 2.8) in Charavet studies[31, 32, 41], and (3± 2) in Yavuz study [43]. However, Gibreal et al.[39] reported that there were no

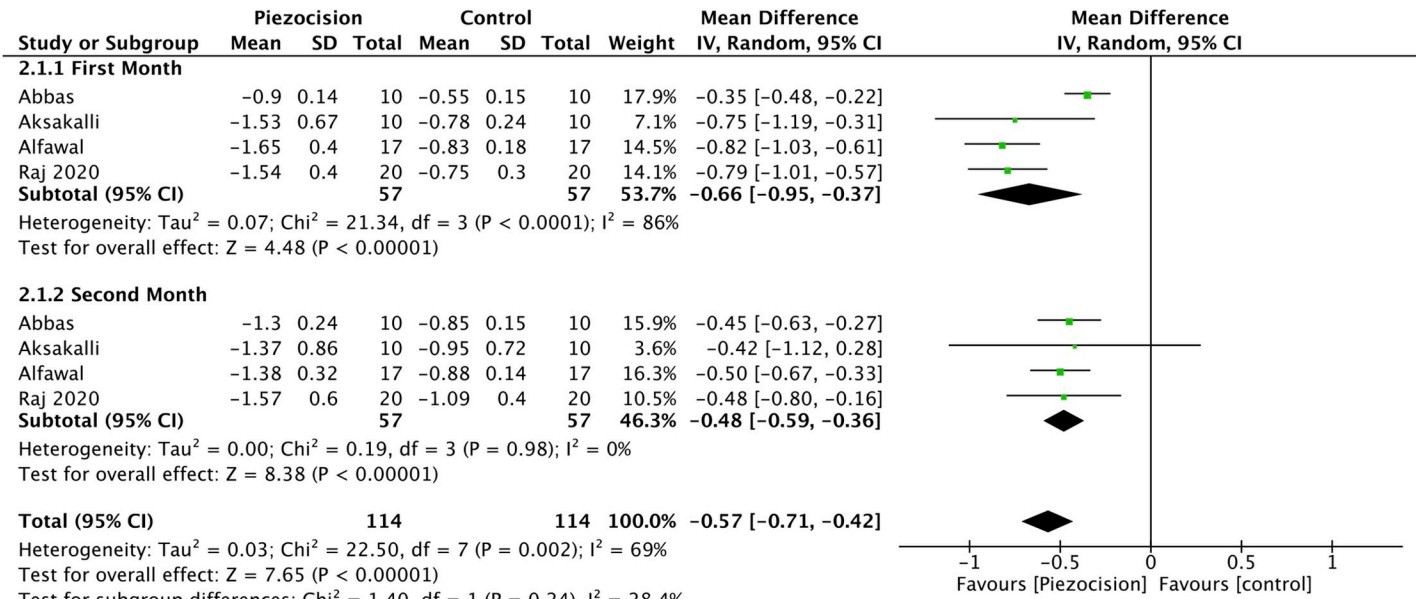

**Fig 3. Forest plot for canine retraction rate (mm/month) between piezocision and control groups for the first two months after surgery with subgroups for the first and second months.**

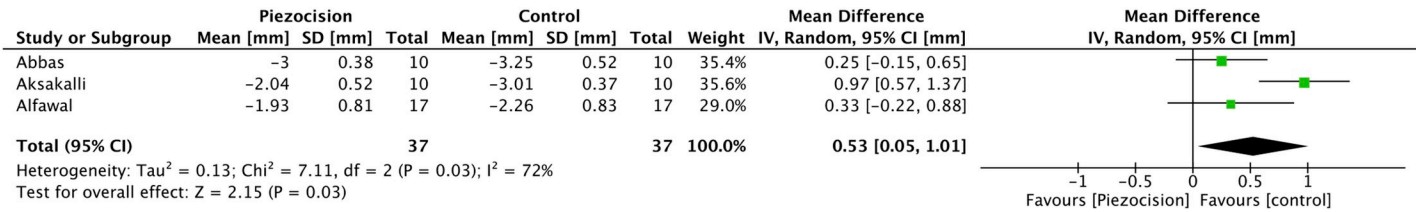

**Fig 4. Forest plot showing anchorage loss differences between piezocision and control groups in canine retraction.**

statistically significant differences regarding pain, discomfort, jaw movement limitations between the piezocision and control group in the first and seventh days after onset of the treatment.

**Additional analysis.** Sensitivity analysis was not possible according to the insufficient studies for each outcome.

**Risk of bias across studies.** GRADE approach[44] was used to rank the quality of the body of evidence. The qualities of the evidence were graded as low (Table 3) because of the high risk of bias within studies in more than one domain, the inconsistency in delivering the intervention, and the small sample sizes in the included studies.

## Discussion

### Summary of evidence

Our meta-analysis showed that piezocision increases the canine retraction rate by 0.57 mm per month for the initial two months after the surgical intervention. There was also a less total molar anchorage loss of 0.53 mm by piezocision. These findings demonstrate the effectiveness of the piezocision procedure in accelerating the canine retraction rate with a statistically significant difference. Similar results regarding accelerating tooth movement were made in other studies [12, 16, 17, 36, 45–47] but differed in the effect size. Interestingly, Fu et al. [17] reported similar results in their meta-analysis, but they included five different surgical interventions, one of them was in combination with laser and pooled heterogeneous methodological and clinical studies in the same meta-analysis. Also, they pooled the 2-weeks movement rate with the monthly movement rate in the same meta-analysis to get the monthly tooth movement rate. Also, Viwattanatipa and Charnchairerk [18] study found five trials that studied effect corticotomy or piezocision on tooth movement, and they included only two studies of our included studies which were done prior to 2018 (Table 4). In addition, there are some doubts about the interpretation of the results by the authors in that meta-analysis as they reported that the surgical intervention could increase the movement rate up to four-fold. However, the present meta-analysis was done regardless of age variation among studies, as there was no significant effect of age variation on the treatment duration in the Mavreas study.[2] However, the

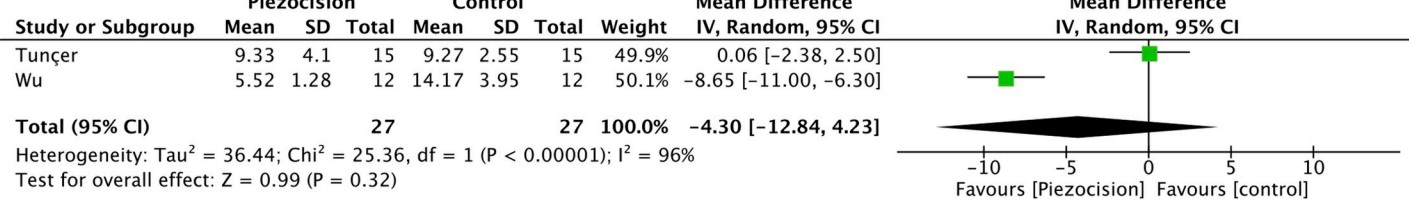

**Fig 5. Forest plot demonstrates the overall en-masse retraction duration differences between piezocision and control groups by months.**

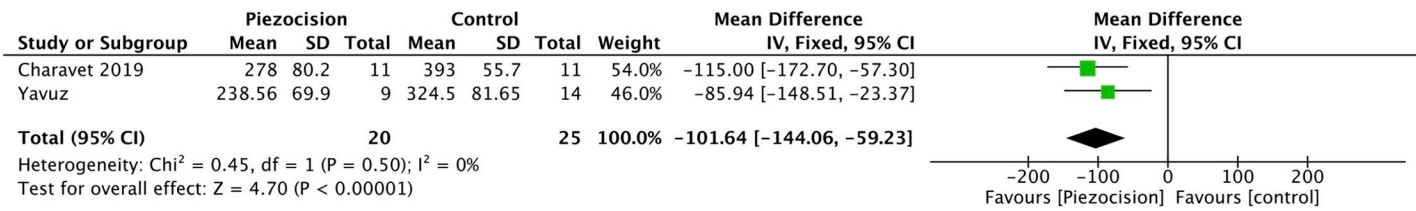

**Fig 6. Forest plot demonstrating piezocision effects on overall treatment duration by months.**

pooled trials [33, 35–37] for this outcome are split-mouth design studies which mostly neglect the correlation between the two sides of mouth during sample size calculation and statistical analysis, and neglect the baseline characteristics of the two sides during randomization process.[48, 49]

The lowest canine retraction rate by piezocision was reported by Alfawal et al.[33] who used a 0.019× 0.025- inch S.S archwire for canine retraction versus 0.016×0.022- inch S.S arch-wire used in the other studies.[35, 36] It is a well-known fact that the archwires size can affect sliding of teeth during space closure.[50]

The highest canine retraction rate (MD; 0.76 mm /month) was reported by Aksakalli [35] but with a very wide confidence interval of 95% (0.19–1.34 mm / month). This wide confidence interval lowers the precision in the point of estimate and increases uncertainty about the intervention estimate in that study.

The acceleration was not significantly higher in the first month after piezocision than the second month (Fig 3). Wilcko et al.[51] reported that RAP reaches its peak between the 3rd and 11th weeks after the bone injury. Only Alfawal et al.[33] revealed that the acceleration was higher in the first month. Otherwise, Abbas et al.[36] reported a higher acceleration in the third month. Raj et al. [37]found that the acceleration increased in the second month and decreased in the third month.

Uribeet al.[29] reported that there was no statistical difference in the alignment time in the piezocision group and control group, which was contradictory to other studies.[32, 43] As the depth of piezocision was 1 mm in the Uribe study[29] versus 3mm for other studies.[29, 32, 43] So, it can be suggested that accelerated tooth movement needs at least a 3 mm penetration depth of the piezocision cuts to get the desirable intended effect. Gibreal et al.[15] concluded that the overall time of alignment in the piezocision group was 59% less than the control group. Although, they demonstrated in their data that alignment was done in more than two months for the two groups, they reported that the alignment time was less than two months in piezocision group, which lead to conflict in their findings.

The effect of piezocision is still unclear in en-masse retraction. The total duration of retraction was shorter but not statistically significant in our meta-analysis (Fig 5). Tuncer et al.[38] found no statistically significant difference in duration of en- masse retraction between control and piezocision groups, while Wu et al.[42] reported a shorter period in the piezocision group. However, there are substantial differences in the study design between these two studies. Interestingly, Wu et al.[42] conducted a clinical trial without randomization, and this may raise the selection bias, also they raised a full thickness flap using bone graft material to cover the bone, and this may have increased the inflammatory response. However, the severity and the type of malocclusion play a role in tooth movement as suggested by Mavreas[2]. Tuncer et al.[38] treated class I or class II patients while Wu et al.[42] treated presurgical class III patients. Also, Wu et al.[42] did not mention the magnitude of the applied force.

The present meta-analysis suggests that piezocision was effective in minimizing the overall duration of orthodontic treatment. It decreased the overall treatment time by more than three

months (Fig 6). This is statistically significant, but we cannot say if it would be clinically significant, as the mean treatment duration varies between 19–34 months.[2]

For maxillary incisors' retraction, Al-Imam et al.[40] found a statistically significant increase in the rate of retraction. According to this study[40] palatal and labial piezocision decreased the time of incisors' retraction by 3 weeks when compared to incisors' retraction without piezocision. That was a small difference if we consider the whole treatment time.[2]

Every treatment has benefits and harms. The reasonable interpretation of less root resorption in the piezocision group[36] is the shorter treatment duration and possibly the change in the bony nature of the surgical site, which is demonstrated in the Segal study[52]. Two studies [31, 32] in this review found similar apical root resorption between the two groups, although these two studies[31, 32] found a significant difference in the treatment duration, while one study[30] showed less root resorption in the piezocision group. Although those two studies reported a significant difference in the treatment duration and have the same active treatment time in the two groups. In contrast, Patterson et al.[53] reported the iatrogenic effect of piezocision procedure and root harms for five patients during surgery, and they found volumetric root resorption in piezocision side. However, Al-Imam et al.[34] reported acute palatal postsurgical inflammation case and they interpreted that it was because of poor oral hygiene, although their criteria comprised the good oral hygiene patients and they prescribed them antibiotics, which suggests that the surgery from the palatal side has more potential risks than when performed on the buccal side.

The periodontal parameters were stable and similar between groups without the probability of recession risk in studies. [30, 32, 41] The remaining scars in the surgical group may cause an esthetic problem for patients with a high smile line, but this is unpredictable. Pain levels were varied from mild [42]to high[31, 41]the day after surgery and decreased in the next seven days. The differences in pain records between studies might be related to age differences between studies as Charavet et al. [31, 41] had patients in their mid-thirties who felt more pain versus 13–19 years old young patients in Yavuz study [43] who felt less pain after surgery. The reported age variation can increase pain experience and may have an effect on the main PROMs psychometric properties.[54]

Previous systematic reviews [12, 19] assessed a number of the studies included in this review[29, 31, 36] as having an unclear risk of bias, but they were assessed as having a high risk of bias in our review, as we have used the last version of the Cochrane Risk of Bias tool (ROB.2), which is more comprehensive and more critical than the original ROB tool.[55]

## Limitations and strength

Although our study was conducted with rigorous methodology according to the Cochrane and PRISMA guidelines, there were still some limitations. First, the included RCTs and CCTs were low quality and had a low sample size and suffered from a lack of randomization and assessor's blinding.

Many studies failed to mention the effect of piezocision on the overall treatment time and did not study the relation between the depth of corticision and the acceleration of tooth movement. Also, none of the included studies reported the harm or iatrogenic effect of piezocision on the roots of the teeth.

Interestingly, our decision was to include one surgical intervention in the meta-analysis to decrease the clinical heterogeneity. However, the statistical heterogeneity among studies ramped up the inconsistency about the estimate, and more subgroup analyses were not possible. In contrast, a recent meta-analysis [17]pooled five different interventions and different outcomes in their meta-analysis.

The present meta-analysis has more strength in terms of studies included than other studies [12, 17–19], as much related primary research has emerged in the last two years. Also, these studies were not registered in PROSPERO.

The registration of the protocol in PROSPERO, which reduces bias in conducting the review [17, 56], the use of the latest Cochrane Risk of Bias tool for the intervention trials, and the GRADE assessment of the studies' quality were strong points for this review. Therefore, we can say with a low certainty that piezocision increases the rate of canine retraction by 0.57 mm/month for the first two months and that piezocision can decrease the overall treatment time by more than three months. Our estimate is limited, and the true effect may be substantially different from our estimate.

## Recommendations

We suggest that more high quality RCTs with large sample sizes should be conducted to study the effect of the piezocision in the long term, with its relapse possibility, and the need to define the overall effect on the treatment time with more focus on the iatrogenic effects of piezocision especially root harms, as well as the relationship between the depth of cuts and the intervention's effectiveness.

## Conclusion

The low-quality evidence suggests that piezocision is an effective surgical procedure in accelerating orthodontic tooth movement, but we should take into account that this effect is clinically small and transient for the first three months according to bone remodeling. Moreover, no high quality RCTs with a large sample size have yet been done in order to help in constructing a more solid scientific point of view regarding this intervention.

In our clinical orthodontic practice, we should weigh the cost and limited benefit of this intervention to our patients, as there are some clear adverse effects like pain and scarring following this type of intervention.

## Supporting information

**S1 Checklist. PRISMA 2009 checklist.**
(DOC)

**S1 Table.**
(DOCX)

**S2 Table.**
(DOCX)

**S3 Table.**
(DOCX)

## Author Contributions

**Writing – original draft:** Samer Mheissen, Haris Khan, Shadi Samawi.

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
