## [Decision Letter · Decision Letter 0]

18 Dec 2019

PONE-D-19-31301

Is Piezocision effective in accelerating orthodontic tooth movement: a systematic review and meta-analysis

PLOS ONE

Dear Dr Mheissen,

Thank you for submitting your manuscript to PLOS ONE. After careful consideration, we feel that it has merit but does not fully meet PLOS ONE’s publication criteria as it currently stands. Therefore, we invite you to submit a revised version of the manuscript that addresses the points raised during the review process.

The reviewers had mixed reviews about your submission. The most crucial point was whether another review is needed on an already saturated subject. You will have to careful indicate exactly how this review improves upon previous studies and if it's really justified with new knowledge. Otherwise, even if the review is done to the highest standards, there is little merit in publishing it. Another issue was that some studies might have been missed from the literature search (one reviewer pointed for example a Gibreal et al BMC Oral Health study). The literature search might not be as up-to-date as it could be. RoB 2.0 is not thought for the assessment of non-randomized studies. There are some technical issues with the meta-analysis part (choice of model made on post-hoc heterogeneity reporting; use of only some heterogeneity estimators; lack of predictions---please check with your statistician). These are some of the issues that need to be addressed before this submission can be further assessed for eligibility.

We would appreciate receiving your revised manuscript by Jan 31 2020 11:59PM. To enhance the reproducibility of your results, we recommend that if applicable you deposit your laboratory protocols in protocols.io, where a protocol can be assigned its own identifier (DOI) such that it can be cited independently in the future. For instructions see: http://journals.plos.org/plosone/s/submission-guidelines#loc-laboratory-protocols

We look forward to receiving your revised manuscript.

Kind regards,

Spyridon N. Papageorgiou, DDS, Dr Med Dent

Academic Editor

PLOS ONE

Journal Requirements:

2. Please amend your manuscript to include your abstract after the title page.

4. Thank you for stating the following in the Financial Disclosure section:

'No funding'

We note that one or more of the authors are employed by a commercial company: Private Practice, Jordan

Reviewers' comments:

Reviewer's Responses to Questions

**Comments to the Author**

1. Is the manuscript technically sound, and do the data support the conclusions?

Reviewer #1: Partly

Reviewer #2: Yes

Reviewer #3: Partly

Reviewer #4: Partly

Reviewer #5: Yes

Reviewer #6: No

2. Has the statistical analysis been performed appropriately and rigorously? 

Reviewer #1: Yes

Reviewer #2: Yes

Reviewer #3: Yes

Reviewer #4: No

Reviewer #5: Yes

Reviewer #6: No

3. Have the authors made all data underlying the findings in their manuscript fully available?

Reviewer #1: Yes

Reviewer #2: Yes

Reviewer #3: Yes

Reviewer #4: Yes

Reviewer #5: No

Reviewer #6: No

4. Is the manuscript presented in an intelligible fashion and written in standard English?

Reviewer #1: Yes

Reviewer #2: Yes

Reviewer #3: Yes

Reviewer #4: Yes

Reviewer #5: Yes

Reviewer #6: No

5. Review Comments to the Author

Reviewer #1: Firstly, I thought that it was great to see a systematic review come up with some positive findings that were clinically relevant. However; Angle Orthodontist published a systematic review titled ‘Efficacy of piezocision on accelerating orthodontic tooth movement:A systematic review.’ Doi: 10.2319/01191-751.1 in 2017. Also, Journal of Clinical Experimental Dentistry published a systematic review ‘Effects of piezocision in orthodontic tooth movement: A systematic review of comparative studies.’ doi:10.4317/jced.56328 in 2019. These investigators came to similar conclusions in presented study. When I looked closely, I found that the Angle systematic review included four eligible studies, and Journal of Clinical Experimental Dentistry included eleven papers however the present study has more trials (12 papers). I leave it to editor whether to publish.

Reviewer #2: Please see attached file.

The authors used systematic review and meta-analysis to obtain the following information.

1. Canine retraction rate (mm/month) between piezocision and

control groups

2. Canine retraction rate (mm/month) between the first and second months

for the canine retraction rate after surgery.

3. Overall canine retraction rate (mm) between piezocision and

control groups

4. Anchorage loss differences between piezocision and control groups.

5. Overall en masse retraction duration differences between piezocision

and control groups (Fig 7) ???

6. En masse retraction overall treatment duration differences between piezocision and control groups . (Fig 8)

Authors made very good papers evaluation in Discussion

Introduction

1. Weak introduction writing

Author should emphasize more on past Piezocision research

- What were already known?

- What were not known?

2. The objective written was different from the methodology and results.

While only canine retraction was mentioned in the objective, the methodology and results also showed Anterior retraction.

“The recent systematic review and meta-analysis aim to critically appraise the

available evidence regarding the effectiveness of piezocision in accelerating

canine retraction in the first two months after piezocision, the effects of

piezocision on treatment duration”

3. What is the Rationale of this study? This should be stated.

MATERIAL AND METHODS

1. Suggestion to delete

Comparison: No adjunct procedures for accelerating tooth movement.

2. Exclusion criteria:

- Should cross sectional, cohort studies be comparative studies?

“Non-comparative studies such as cross sectional, cohort studies, case

series, or case reports”

3. Risk of bias

Figure 2 shows 2 figures: should this be classified as a) / b)

Please briefly explain how to get figure 2. Upper one

4. Twelve papers included were heterogenous in terms of orthodontic treatment and outcome measured.

For example

- Ref 22. Ma et al.: orthodontic traction of impacted mandibular third molars

- Ref 28. Tuncer: En-masse retraction

- Ref 24. Gibreal: crowded lower anterior teeth

5. Is it sufficient to conduct metaanalysis with only 2 published papers in Fig 7/ Fig 8?

6. Fig 7: The unit of outcomes used is not clear. Is it distance in mm.?

Reviewer #3: The topic of the article is interesting and precious for academics as well as clinicians.

There are several issues that need to be addressed in order to considerably improve the quality of the article, notably the justification of a new systematic review of the piezocision topic and on the conclusions section.

Some comments:

Currently, systematic reviews about piezocision are provided in the literature. The authors must justify in which way their systematic review provides us with new information about the piezocision procedure. If the authors cannot justify this, the paper must be rejected.

Figure:

Figure 2: In the study of Charavet et al., the randomization process was clearly explained in an appendix section. Please read this part.

Introduction section:

The piezocision protocol involved at least 3 mm deep (NOT 1 mm) incisions to be effective in accelerating the orthodontic tooth movement. Please correct this mention.

Furthermore, the authors must explain the surgical protocol of piezocision in detail. It has not been clearly explained in the introduction.

Then, piezocision is always without a flap. Why do the authors say “mostly involves flapless corticotomies”?

There was some errors or lack of precision regarding the piezocision procedure.

Concerning the RAP, the authors should cite the most recent publication, notably in animal studies.

Why do the authors speak about the “potential complication of root damage”? Are there some relevant publications about this? Also, the authors should cite relevant publications concerning surgical guides.

Materials and methods section

Did the two reviewers do some meeting training to fully understand the protocol?

Did you have a training meeting between the two reviewers?

There is a minimum of impact factor for the choice of the orthodontic journals? Please be more precise.

Discussion section :

The discussion should mention the effect of the piezocision according to the two type of designs:

(1) The effect of piezocision in the case of canine retraction

(2) The effect of piezocision in the case of overcrowding treatment

The part "recommandations" was not relevant. Root resorption was investigated in at least 2 previous RCTs. The relationship with the depth of cuts and the effectiveness was fully explained and demonstrated : at least 3 mm depth to be effective. A large sample size is not a relevant criteria compared to the simple size calculation. Please modify accordingly.

It would be much better to create a section "guidelines for surgery and orthodontics" section, according to your results.

Conclusion section :

This section must be considerably improved.

Why do the authors claim “low-quality evidence”?

And why “no high quality RCTs with a large sample size”? In a RCT you have to respect your simple size calculation. If it’s the case, you don’t need a large simple size to demonstrate more effectively the effect of a procedure. This sentence is also not well-formulated.

Why use the term of “small”? It’s not scientific and involves a wrong interpretation of the piezocision kinetic effect.

Reviewer #4: The authors set out to analyze the studies and produce a systematic review and meta analysis on piezocision and its effect on acceleration of the rate of tooth movement. The following concerns were found with this study:

1) In the intervention, it should be listed that a flapless approach to access the bone as this is critical for the “ non-invasive piezocision method”. The mucoperisteal flap procedure has been reported as a major factor that could induce the increase in the rate of tooth movement. Studies using a flap (such as Wu study) should had been excluded.

1) It is not clear what specifically was the high risk of bias in the randomization process for 3 studies what were the concerns on other 5. What were the specific problems for these studies in the randomization method.

2) What were the specific problems regarding measurement of outcome?

3) It is not clear why was the data the data for those studies evaluating the alignment phase not pooled and analyzed separately.

4) Figure 4 is misleading for the Aksakalli study. In the second month, they did not have 2.9 mm of canine movement, this was the total movement. The amount of movement was 1.3mm.

5) Figure 5 has the same problem for the total movement over 2 months. The total movement was 2.9 and not 4.43 (the authors are wrongly adding the total displacement at the second month with the first month displacement). The same problem is observed for the control group for this study. The forest plot is inaccurate.

6) Figure 5 describes overall retraction rate. It is not clear what is the endpoint for the different studies. This review’s objective was to measure only the first 2 months.

7) As stated above, Wu study should have been excluded as it has a full mucoperioteal flap to perform the alveolar decortication with a piezotome. It has been suggested that the flap may be responsible to certain extent to the acceleration in the rate of tooth movement. Additionally, the authors set in the exclusion criteria “participants undergoing orthognathic surgery”. All the patients in this study were to receive orthognathic surgery.

8) Table III summarizes the canine retraction and states that was based on 4 RCT. The authors have only reported 3 RCTs to extract the data.

Minor

A reference should be used after this statement: “Radiographic metal guides placed on

archwires have been advocated to avoid this complication.”

Participants eligibility criteria should also include patients taking medications that can affect tooth movement.

Multiple grammar mistakes throughout the document.

Overall my enthusiasm for this systematic review/meta-analysis is very low. There are multiple problems on the inclusion of studies and the reporting of the review as illustrated above. Based on what has been collected, it is hard to abstract any useful information. I would recommend the submission to another journal.

Reviewer #5: In this article the authors assessed in a systematic way the effects of piezocision as an adjunct procedure to accelerate orthodontic tooth movement.

They followed the methodology as described in the Cochrane Handbook for evaluating of both for canine retraction for extraction space closure and treatment time duration.

I would recommend in the “Discussion” section and in the part that root resorption of the included studies is discussed to add the limitations of assessing root resorption with radiography. Given that there are studies that assessed root resorption of piezocision and other minimally invasive procedures, such as osteoperforations using micro-CT, it would be beneficial for the readers to have these references and an argument in the discussion about this factor.

The “supplemental 2” has not been successfully uploaded.

There is no information about the excluded studies.

It is advisable to include a list with the stage of exclusion (title, abstract/full text) of all remained studies after removal of duplicates.

In the “full text” stage of exclusion, it is recommended to include the reasons of exclusion for each of the excluded studies.

Please also revise for grammar/syntax errors. There were few in the manuscript.

Reviewer #6: The following points should be taken into account in any further re-submission:

The English language should be improved in the whole manuscript. Many grammatical errors should be corrected. Many phrase should be improved. Some areas require additional care in scientific writing up.

The first reports about the use of piezosurgery are not published by Dibart et al. Actually, in 2007, Vercelotti and Podesta introduced the use of piezosurgery, instead of burs, in conjunction with the conventional flap elevations

to create an environment conducive to rapid tooth movement. Therefore, the authors are advised to change the introduction accordingly.

Several papers should have been added to the final list of accepted and included studies. For example, Gibreal et al published a paper in the BMC Oral Health Journal in April 2019 titled "Evaluation of the levels of pain and discomfort of piezocision-assisted flapless corticotomy when treating severely crowded lower anterior teeth: a single-center, randomized controlled clinical trial". However, this paper was not included in this systematic review although the authors claim that they searched the literature till April 2019.

Exclusion criteria should be expanded to cover a wide spectrum of papers that should not be included into this systematic review, e.g. 'corticision' in its different possible applications.

6. PLOS authors have the option to publish the peer review history of their article (what does this mean?). If published, this will include your full peer review and any attached files.

Reviewer #1: No

Reviewer #2: No

Reviewer #3: No

Reviewer #4: No

Reviewer #5: No

Reviewer #6: No

---

## [Author Response · Author response to Decision Letter 0]

2 Jan 2020

PONE-D-19-31301

Is Piezocision effective in accelerating orthodontic tooth movement: a systematic review and meta-analysis

PLOS ONE

Dear Editor Dr. Papageorgiou

Thank you for your kind revision letter. We write this letter to respond to each point raised by the academic editor and reviewers.

Academic editor:

The reviewers had mixed reviews about your submission. The most crucial point was whether another review is needed on an already saturated subject. You will have to careful indicate exactly how this review improves upon previous studies and if it's really justified with new knowledge. Otherwise, even if the review is done to the highest standards, there is little merit in publishing it. Another issue was that some studies might have been missed from the literature search (one reviewer pointed for example a Gibreal et al BMC Oral Health study). The literature search might not be as up-to-date as it could be. RoB 2.0 is not thought for the assessment of non-randomized studies. There are some technical issues with the meta-analysis part (choice of model made on post-hoc heterogeneity reporting; use of only some heterogeneity estimators; lack of predictions---please check with your statistician). These are some of the issues that need to be addressed before this submission can be further assessed for eligibility.

Authors response:

We started this review from march 2019 and the delay happened due to PROSPERO registration, so two systematic reviews[1, 2] were published in the meanwhile when we were in the registration process. Both of the published reviews are unregistered. One of the aim of a registered review is to decrease the bias while conducting the review.

 Figueiredo review [1]has a publishing review process for 20 days and it is a qualitative systematic review. Fu review[2] included five interventions and one of them was decortication with laser and they have a major mistake in meta-analysis.

So, our review is the only registered review in PROSPERO, included 15 records about piezocision which is more than any previous review, and has a quantitative analysis. The inclusion of more studies may will not lead to different results but can increase the precision and reduce the random error. [3]

Gibreal et al.[4] study was published after our search date (2-10-2019) but we included it with two other RCTs in the revised manuscript.

For the CCTs, we reassessed them by ROBINS. However, we know that RoB 2.0 is not thought for the assessment of non-randomized studies, but we emailed Julian Higgins (the editor of Cochrane handbook) and this is his response; 

“Dear Samer,

There is an area of overlap here, and I would say that the decision depends on:

(i) how similar to a randomized trial your clinical trials appear to be (e.g. if it looks as if the could have been randomized then I’d use the RoB 2 tool and answer ‘no information’ about the randomization methods).

(ii) what other studies you have in your review, if any (e.g. if all the other studies are randomized trials, I’d use RoB 2; if all the others are non-randomized, I’d use ROBINS-I).”

- For the meta-analysis, we did more consultation and changed it.

Reviewer #1: Firstly, I thought that it was great to see a systematic review come up with some positive findings that were clinically relevant. However; Angle Orthodontist published a systematic review titled ‘Efficacy of piezocision on accelerating orthodontic tooth movement: A systematic review.’ Doi: 10.2319/01191-751.1 in 2017. Also, Journal of Clinical Experimental Dentistry published a systematic review ‘Effects of piezocision in orthodontic tooth movement: A systematic review of comparative studies.’ doi:10.4317/jced.56328 in 2019. These investigators came to similar conclusions in presented study. When I looked closely, I found that the Angle systematic review included four eligible studies, and Journal of Clinical Experimental Dentistry included eleven papers however the present study has more trials (12 papers). I leave it to editor whether to publish.

Authors response:

We appreciate your comments and have the same concerns, but the two mentioned reviews are qualitative systematic reviews [1, 5] not quantitative systematic reviews. In contrast we increased our included studies to 15 studies, and we have done a quantitative systematic review. Also, the recent published review[1] made a mistake in treating two included studies as CCT while they are RCT.

Reviewer #2: Please see attached file.

The authors used systematic review and meta-analysis to obtain the following information.

1. Canine retraction rate (mm/month) between piezocision and

control groups

2. Canine retraction rate (mm/month) between the first and second months

for the canine retraction rate after surgery.

3. Overall canine retraction rate (mm) between piezocision and

control groups

4. Anchorage loss differences between piezocision and control groups.

5. Overall en masse retraction duration differences between piezocision

and control groups (Fig 7) ???

6. En masse retraction overall treatment duration differences between piezocision and control groups . (Fig 8)

Authors made very good papers evaluation in Discussion

Introduction

1. Weak introduction writing

Author should emphasize more on past Piezocision research

- What were already known?

- What were not known?

2. The objective written was different from the methodology and results.

While only canine retraction was mentioned in the objective, the methodology and results also showed Anterior retraction.

“The recent systematic review and meta-analysis aim to critically appraise the

available evidence regarding the effectiveness of piezocision in accelerating

canine retraction in the first two months after piezocision, the effects of

piezocision on treatment duration”

3. What is the Rationale of this study? This should be stated.

MATERIAL AND METHODS

1. Suggestion to delete

Comparison: No adjunct procedures for accelerating tooth movement.

2. Exclusion criteria:

- Should cross sectional, cohort studies be comparative studies?

“Non-comparative studies such as cross sectional, cohort studies, case

series, or case reports”

3. Risk of bias

Figure 2 shows 2 figures: should this be classified as a) / b)

Please briefly explain how to get figure 2. Upper one

4. Twelve papers included were heterogenous in terms of orthodontic treatment and outcome measured.

For example

- Ref 22. Ma et al.: orthodontic traction of impacted mandibular third molars

- Ref 28. Tuncer: En-masse retraction

- Ref 24. Gibreal: crowded lower anterior teeth

5. Is it sufficient to conduct metaanalysis with only 2 published papers in Fig 7/ Fig 8?

6. Fig 7: The unit of outcomes used is not clear. Is it distance in mm.?

Authors response:

Thank you for your interesting comments.

1. For the Introduction, we made some amendments in it.

2. The objective was in the protocol but after we finished the inclusion studies, we found segmental retraction after premolars extraction and this is relevant to canine retraction.

3. we wrote in the introduction the rational (this is for pure quantitative systematic review)

For the Methods:

1. This is the control or the comparison in PICO question why we should delete?

2. This part is modified according to reviewer suggestion

3. This part is modified according to reviewer suggestion

Now we used Risk of bias website to get them.

4. For the meta-analysis, we separated them under many categories.

5. Jada et al[3] mentioned that.

6. The unit there is days 

Reviewer #3: The topic of the article is interesting and precious for academics as well as clinicians.

There are several issues that need to be addressed in order to considerably improve the quality of the article, notably the justification of a new systematic review of the piezocision topic and on the conclusions section.

Some comments:

Currently, systematic reviews about piezocision are provided in the literature. The authors must justify in which way their systematic review provides us with new information about the piezocision procedure. If the authors cannot justify this, the paper must be rejected.

Figure:

Figure 2: In the study of Charavet et al., the randomization process was clearly explained in an appendix section. Please read this part.

Introduction section:

The piezocision protocol involved at least 3 mm deep (NOT 1 mm) incisions to be effective in accelerating the orthodontic tooth movement. Please correct this mention.

Furthermore, the authors must explain the surgical protocol of piezocision in detail. It has not been clearly explained in the introduction.

Then, piezocision is always without a flap. Why do the authors say “mostly involves flapless corticotomies”?

There was some errors or lack of precision regarding the piezocision procedure.

Concerning the RAP, the authors should cite the most recent publication, notably in animal studies.

Why do the authors speak about the “potential complication of root damage”? Are there some relevant publications about this? Also, the authors should cite relevant publications concerning surgical guides.

Materials and methods section

Did the two reviewers do some meeting training to fully understand the protocol?

Did you have a training meeting between the two reviewers?

There is a minimum of impact factor for the choice of the orthodontic journals? Please be more precise.

Discussion section :

The discussion should mention the effect of the piezocision according to the two type of designs:

(1) The effect of piezocision in the case of canine retraction

(2) The effect of piezocision in the case of overcrowding treatment

The part "recommandations" was not relevant. Root resorption was investigated in at least 2 previous RCTs. The relationship with the depth of cuts and the effectiveness was fully explained and demonstrated : at least 3 mm depth to be effective. A large sample size is not a relevant criteria compared to the simple size calculation. Please modify accordingly.

It would be much better to create a section "guidelines for surgery and orthodontics" section, according to your results.

Conclusion section :

This section must be considerably improved.

Why do the authors claim “low-quality evidence”?

And why “no high quality RCTs with a large sample size”? In a RCT you have to respect your simple size calculation. If it’s the case, you don’t need a large simple size to demonstrate more effectively the effect of a procedure. This sentence is also not well-formulated.

Why use the term of “small”? It’s not scientific and involves a wrong interpretation of the piezocision kinetic effect.

Authors response:

Thank you for your positive comments and your crucial points.

We started this review from march and the delay happened from the PROSPERO registration, so two systematic reviews[1, 2] were published when we in the review process. None of them registered his review in PROSPERO. Figueiredo review [1]has a publishing review process for 20 days and it is a qualitative systematic review. Fu review included five interventions on of them was decortication with laser and they have a major mistake in meta analysis.

So, our review is the only registered review in PROSPERO, included 15 records about piezocision which is more than any previous review, and has a quantitative analysis. The inclusion of more studies may will not lead to different results but can increase the precision and reduce the random error. [3]

Figure2; We agree with you but the ROB2 tool has three parts and the last part is (Did baseline differences between intervention groups suggest a problem with the randomization process?) and there are baseline differences between groups.

Introduction section:

The 1 mm deep cuts were mentioned by Uribe et al.[6], so we have written in the introduction the deep is between 1-3 mm.

The introduction was modified for these comments and for more citations. 

Wu et al.[7] used piezocision with a flap.

Patterson et al. [8] studied iatrogenic effects on roots, and we added this in the discussion.

Materials and methods section

There was a meeting every two days to assess the studies, and we did primarily assessment for one study to have a consensus for the three authors then we assessed all the papers.

Discussion section:

We have mentioned 5 topics related to piesocision in this review and we think if made more subtitles there will be more confusion for the reader.

Till now we know from the evidence the 3 mm depth works and the 1 mm is not working but we don’t know about the other depths.

We think for guidelines for orthodontist and surgeons we need more than this review.

We made some amendments in the conclusion.

We found the RCTs with a high risk of bias using ROB2 and this downgraded the quality of the RCTs, also the heterogeneity and the inconsistency between the results.

Sample size calculation depends on the effect size and if we increase the effects size the sample will decrease. So, in the included studies the effect size was high which decreased the sample size. 

Reviewer #4: The authors set out to analyze the studies and produce a systematic review and meta-analysis on piezocision and its effect on acceleration of the rate of tooth movement. The following concerns were found with this study:

1) In the intervention, it should be listed that a flapless approach to access the bone as this is critical for the “ non-invasive piezocision method”. The mucoperisteal flap procedure has been reported as a major factor that could induce the increase in the rate of tooth movement. Studies using a flap (such as Wu study) should had been excluded.

2) It is not clear what specifically was the high risk of bias in the randomization process for 3 studies what were the concerns on other 5. What were the specific problems for these studies in the randomization method.

3) What were the specific problems regarding measurement of outcome?

4) It is not clear why was the data the data for those studies evaluating the alignment phase not pooled and analyzed separately.

5) Figure 4 is misleading for the Aksakalli study. In the second month, they did not have 2.9 mm of canine movement, this was the total movement. The amount of movement was 1.3mm.

6) Figure 5 has the same problem for the total movement over 2 months. The total movement was 2.9 and not 4.43 (the authors are wrongly adding the total displacement at the second month with the first month displacement). The same problem is observed for the control group for this study. The forest plot is inaccurate.

7) Figure 5 describes overall retraction rate. It is not clear what is the endpoint for the different studies. This review’s objective was to measure only the first 2 months.

8) As stated above, Wu study should have been excluded as it has a full mucoperioteal flap to perform the alveolar decortication with a piezotome. It has been suggested that the flap may be responsible to certain extent to the acceleration in the rate of tooth movement. Additionally, the authors set in the exclusion criteria “participants undergoing orthognathic surgery”. All the patients in this study were to receive orthognathic surgery.

9) Table III summarizes the canine retraction and states that was based on 4 RCT. The authors have only reported 3 RCTs to extract the data.

Minor

A reference should be used after this statement: “Radiographic metal guides placed on

archwires have been advocated to avoid this complication.”

Participants eligibility criteria should also include patients taking medications that can affect tooth movement.

Multiple grammar mistakes throughout the document.

Overall my enthusiasm for this systematic review/meta-analysis is very low. There are multiple problems on the inclusion of studies and the reporting of the review as illustrated above. Based on what has been collected, it is hard to abstract any useful information. I would recommend the submission to another journal.

Authors response:

1) This have been discussed in the discussion that why the studies were classified into a certain group on the basis of risk of bais tool.

2) We agree with you but the ROB2 tool has three parts and the last part is (Did baseline differences between intervention groups suggest a problem with the randomization process?) 

3) The method of measurement if done with more precise and rigorous way

4) Because one did the alignment before the surgery and one after the surgery.

5) We corrected it

6) We deleted it

7) We deleted it

8) Agree with you, but the orthognathic surgery is done after the piezocision

9) We corrected it

We used a reference for this guidance.

we modified the inclusion criteria as you suggested

Reviewer #5: In this article the authors assessed in a systematic way the effects of piezocision as an adjunct procedure to accelerate orthodontic tooth movement.

They followed the methodology as described in the Cochrane Handbook for evaluating of both for canine retraction for extraction space closure and treatment time duration.

I would recommend in the “Discussion” section and in the part that root resorption of the included studies is discussed to add the limitations of assessing root resorption with radiography. Given that there are studies that assessed root resorption of piezocision and other minimally invasive procedures, such as osteoperforations using micro-CT, it would be beneficial for the readers to have these references and an argument in the discussion about this factor.

The “supplemental 2” has not been successfully uploaded.

There is no information about the excluded studies.

It is advisable to include a list with the stage of exclusion (title, abstract/full text) of all remained studies after removal of duplicates.

In the “full text” stage of exclusion, it is recommended to include the reasons of exclusion for each of the excluded studies.

Please also revise for grammar/syntax errors. There were few in the manuscript.

Response:

Thank you for your valuable comments. We discussed a new reference with micro-CT as you suggested, and we will re-submit the supplemental 2 for excluded studies with the reason of exclusion.

We also revised the grammar mistakes.

Reviewer #6: The following points should be taken into account in any further re-submission:

The English language should be improved in the whole manuscript. Many grammatical errors should be corrected. Many phrase should be improved. Some areas require additional care in scientific writing up.

The first reports about the use of piezosurgery are not published by Dibart et al. Actually, in 2007, Vercelotti and Podesta introduced the use of piezosurgery, instead of burs, in conjunction with the conventional flap elevations

to create an environment conducive to rapid tooth movement. Therefore, the authors are advised to change the introduction accordingly.

Several papers should have been added to the final list of accepted and included studies. For example, Gibreal et al published a paper in the BMC Oral Health Journal in April 2019 titled "Evaluation of the levels of pain and discomfort of piezocision-assisted flapless corticotomy when treating severely crowded lower anterior teeth: a single-center, randomized controlled clinical trial". However, this paper was not included in this systematic review although the authors claim that they searched the literature till April 2019.

Exclusion criteria should be expanded to cover a wide spectrum of papers that should not be included into this systematic review, e.g. 'corticision' in its different possible applications.

Response:

We appreciate your comments. So, we did a proof reading for the manuscript. Also, we modified the introduction in regard to reports about the use of piezosurgery.

The inclusion of more studies may will not lead to different results but can increase the precision and reduce the random error. [3] Hwoever, Gibreal et al.[4] study was published after our search date (2-10-2019) but we included it with two other RCTs in the revised manuscript. Now we have included 15 records about piezocision which is more than any previous review,

1. Figueiredo D-S-F, Houara R-G, Mata-Cid Pinto L-S-d, Diniz A-R, de Araújo Vn-E, Thabane L, et al. Effects of piezocision in orthodontic tooth movement: A systematic review of comparative studies. J Clin Exp Dent 2019.

2. Fu T, Liu S, Zhao H, Cao M, Zhang R. Effectiveness and Safety of Minimally Invasive Orthodontic Tooth Movement Acceleration: A Systematic Review and Meta-analysis. Journal of Dental Research. 2019. doi: 1d0o.i.1o1rg7/71/00.10127270/0304252103948571894817824.

3. Jadad AR, Cook DJ, Browman GP. A guide to interpreting discordant systematic reviews. CAN MED ASSOC J. 1997;156(10).

4. Gibreal O, Hajeer MY, Brad B. Evaluation of the levels of pain and discomfort of piezocision-assisted flapless corticotomy when treating severely crowded lower anterior teeth: a single-center, randomized controlled clinical trial. BMC oral health. 2019;19(1):57. Epub 2019/04/18. doi: 10.1186/s12903-019-0758-9. PubMed PMID: 30991984; PubMed Central PMCID: PMCPMC6469154.

5. Yi J, Xiao J, Li Y, Li X, Zhao Z. Efficacy of piezocision on accelerating orthodontic tooth movement: A systematic review. The Angle orthodontist. 2017;87(4):491-8. Epub 2017/04/22. doi: 10.2319/01191-751.1. PubMed PMID: 28429956.

6. Uribe F, Davoody L, Mehr R, Jayaratne YSN, Almas K, Sobue T, et al. Efficiency of piezotome-corticision assisted orthodontics in alleviating mandibular anterior crowding-a randomized clinical trial. European journal of orthodontics. 2017;39(6):595-600. Epub 2017/04/04. doi: 10.1093/ejo/cjw091. PubMed PMID: 28371882.

7. Wu J, Jiang J-H, Xu L, Liang C, Bai Y, Zou W. A pilot clinical study of Class III surgical patients facilitated by improved accelerated osteogenic orthodontic treatments. The Angle orthodontist. 2015;85(4):616-24. doi: 10.2319/032414-220.1.

8. Patterson BM, Dalci O, Papadopoulou AK, Madukuri S, Mahon J, Petocz P, et al. Effect of piezocision on root resorption associated with orthodontic force: A microcomputed tomography study. American journal of orthodontics and dentofacial orthopedics : official publication of the American Association of Orthodontists, its constituent societies, and the American Board of Orthodontics. 2017;151(1):53-62. Epub 2016/12/28. doi: 10.1016/j.ajodo.2016.06.032. PubMed PMID: 28024782.

---

## [Decision Letter · Decision Letter 1]

10 Feb 2020

PONE-D-19-31301R1

Is Piezocision effective in accelerating orthodontic tooth movement: a systematic review and meta-analysis

PLOS ONE

Dear Dr Mheissen,

Thank you for submitting your manuscript to PLOS ONE. After careful consideration, we feel that it has merit but does not fully meet PLOS ONE’s publication criteria as it currently stands. Therefore, we invite you to submit a revised version of the manuscript that addresses the points raised during the review process.

All reviewers felt that you had done considerable efforts to answer their comments and improve on the qualilty of your manuscript. However, many of them pointed out some old or new issues (most of them minor) that could be taken into consideration.

We would appreciate receiving your revised manuscript by Mar 26 2020 11:59PM. To enhance the reproducibility of your results, we recommend that if applicable you deposit your laboratory protocols in protocols.io, where a protocol can be assigned its own identifier (DOI) such that it can be cited independently in the future. For instructions see: http://journals.plos.org/plosone/s/submission-guidelines#loc-laboratory-protocols

We look forward to receiving your revised manuscript.

Kind regards,

Spyridon N. Papageorgiou, DDS, Dr Med Dent

Academic Editor

PLOS ONE

Reviewers' comments:

Reviewer's Responses to Questions

**Comments to the Author**

1. If the authors have adequately addressed your comments raised in a previous round of review and you feel that this manuscript is now acceptable for publication, you may indicate that here to bypass the “Comments to the Author” section, enter your conflict of interest statement in the “Confidential to Editor” section, and submit your "Accept" recommendation.

Reviewer #1: All comments have been addressed

Reviewer #2: All comments have been addressed

Reviewer #3: All comments have been addressed

Reviewer #5: All comments have been addressed

Reviewer #6: (No Response)

2. Is the manuscript technically sound, and do the data support the conclusions?

Reviewer #1: Yes

Reviewer #2: Yes

Reviewer #3: No

Reviewer #5: Partly

Reviewer #6: No

3. Has the statistical analysis been performed appropriately and rigorously? 

Reviewer #1: Yes

Reviewer #2: Yes

Reviewer #3: Yes

Reviewer #5: Yes

Reviewer #6: No

4. Have the authors made all data underlying the findings in their manuscript fully available?

Reviewer #1: Yes

Reviewer #2: Yes

Reviewer #3: Yes

Reviewer #5: Yes

Reviewer #6: No

5. Is the manuscript presented in an intelligible fashion and written in standard English?

Reviewer #1: Yes

Reviewer #2: Yes

Reviewer #3: Yes

Reviewer #5: Yes

Reviewer #6: No

6. Review Comments to the Author

Reviewer #1: The authors have successfully addressed previously raised comments. Authors reviewed the manuscript as suggested.

Reviewer #2: Authors’ efforts in revising their manuscript is appreciated.

MATERIAL AND METHODS

1. Suggestion to rephrase for clear understanding: PICOS

From: Comparison: No adjunct procedures for accelerating tooth movement.

To: Comparison: Piezocision VS Control

2. Please recheck consistency across tables and figures.

Ma's et al study appears in table II and Fig 2A/ Not appear in Table IV

Because authors' study focus canine / Anterior retraction, does Ma's et al study actually fit authors' criteria?

Should it be included in final full text evaluation?

Ref 30. Ma et al.: orthodontic traction of impacted mandibular third molars

Ma Z, Xu G, Yang C, Xie Q, Shen Y, Zhang S. Efficacy of the technique of

piezoelectric corticotomy for orthodontic traction of impacted mandibular third

molars. Br J Oral Maxillofac Surg. 2015;53(4):326-31. Epub 2015/02/02. doi:

10.1016/j.bjoms.2015.01.002. PubMed PMID: 25638568.

3. Wu et al's study should be noted in discussion about their technique of raising flap.

Ref 39. Wu J, Jiang J-H, Xu L, Liang C, Bai Y, Zou W. A pilot clinical study of Class III

surgical patients facilitated by improved accelerated osteogenic orthodontic

treatments. The Angle Orthodontist. 2015;85(4):616-24. doi: 10.2319/032414-

220.1.

Reviewer #3: Dear authors,

All my questions and comments have not been resolved.

So, first, thanks for doing it.

Abstract (conclusion part)

I don't agree with the "first three months after the surgery".

Please precise according to the type of orthodontic movement.

Please again in the introduction section, it's wrong to say that the surgical incision is performed 1mm to 3 mm deep. According to Dibart et al., who describe for the first time the piezocision procedure, surgical incision must be performed at least of 3 mm. Please correct this.

Materials and Methods

"healthy patients" is not scientific. Please adjust.

Discussion section

Again, please see my previous comments.

The discussion should be mentioned the effect of the piezocision according to the two types of designs : (1) in case of canine retraction and (2) in case of overcrowding treatment

Reviewer #5: Additional comment.

The authors are advised to add in the discussion the problems arising from the split-mouth individual studies and the quality of the data deriving from them, which are further used for the meta-analysis. More specifically, these studies suffer from inappropriate statistical methods for the assessment of their raw data.

As long as the original studies did not use and did not follow the criteria as set by the following references, then the results of the meta-analysis should be interpreted accordingly.

a. B Chung, N Pandis, R W Scherer, D Elbourne. CONSORT Extension for Within-Person Randomized Clinical Trials J Dent Res 2020; 99 (2): 121-124.

b. Nikolaos Pandis , Bryan Chung Roberta W Scherer , Diana Elbourne , Douglas G Altman CONSORT 2010 Statement: Extension Checklist for Reporting Within Person Randomised Trials BMJ 2017, 357, j2835.

It is recommended to modify the discussion according to the above comment.

Reviewer #6: Abstract

1- Objectives did not include the 'patient-reported outcomes' as one of the aims of this systematic review. The authors mentioned only 'adverse effects' which is not similar to 'patient-reported outcomes'.

2- Search Methods: The English language should be improved in this section.

Introduction

3- The justification of the onset of this systematic review should be mentioned here. The authors are advised to put all of the reasons for conducting this SR at the end of the Introduction section. Here, the authors say claim that no previous systematic review has given the pure effect of piezocision although there is the well-known systematic review of Alfawal et al (2016) published in the Progress in Orthodontics journal which has given some conclusions very similar to the given here in this SR. The authors are also encouraged to include this review in their bibliographic list. Another recent publication (systematic review) assessed different methods of acceleration (mainly piezocision) on the en-masse retraction of upper anterior teeth and this SR should also be visited by the authors to update their introduction, justification and bibliographic list. The paper is written by Khlef, Hajeer, et al (2019) and was published in the JCDP (PMID 31058623).

Objective:

4- The given objective in the text is wrong. The authors should mention that this review aimed to evaluate the available evidence regarding tooth movement acceleration using piezocision (and not only canine retraction). When you have a quick at the retrieved papers, you would immediately discover that the included papers covered a wide range of tooth movements and not only canine retraction.

Materials and Methods

5- Participants: The authors should say 'healthy patients with any Class of malocclusion...' and not 'patients without craniofacial anomalies', since the second phrase is vague and its actual positioning lies in the 'Exclusion Criteria' not the 'Inclusion Criteria'.

6- Intervention. Another mistake appeared here. The 'Intervention' should be 'An acceleration method based on piezosurgery' regardless the orientation of the grooves or fissures. Many piezocision-assisted techniques are based on vertical and horizontal movements of the cutting head. Therefore, this criterion should be corrected.

7- Comparison. The authors should say: 'Patients with malocclusion being treated traditionally without any specific method of acceleration'. The English language should be improved although the authors have claimed that the revised version of their paper has been checked and improved.

8- Outcomes. Again the authors are a little bit confused. The main outcome should be tooth movement velocity. They have now papers discussing canine retraction, en-masse retraction, four-upper-incisors retraction, and leveling and alignment of lower anterior teeth. Therefore being confined to only canine retraction is totally wrong.

Exclusion Criteria

9- What is meant by 'Cortication for rapid maxillary expansion'?

10- What is meant by 'Patients taking medications that can affect tooth movement'?

11- According this criterion 'Involving participants undergoing orthognathic surgery', how did the author include the study of Wu et al (2015; PMID: 25347045) in their qualitative and quantitative analysis? According to this criterion, their narrative evaluation should be rephrased again and the related meta-analysis should be omitted. Otherwise, the reviewers are strongly advised to take the results of a recent RCT (under publication) about en-masse retraction written by Khlef et al (2020) that compared flapless corticotomy (by piezocision) with traditional corticotomy (using burs and flap reflection) in terms of tooth velocity and treatment time. Results can be obtained directly from the corresponding author and main supervisor of this research project (Hajeer MY).

12- The authors say 'Non-comparative studies, such as cohort studies'. This is really strange since it is well known that cohort studies can be accompanied with control groups which make them 'comparative studies'. The authors should rephrase this criterion scientifically again.

Information sources

13- How did the author calculate Kappa statistics? Which variables did they include in the Kappa analysis of agreement. Tables are required here as Supplementary Tables.

Data Collection

14- What kind of data types did the author collect according to the prepared data extraction sheets? These should be mentioned here.

Risk of Bias

15- Did the five points appear in the ROB2 tool or ROBINS-I tool? Which points were similar between the two tools? The authors are asked to make the items of each tool clear to the readers.

Summary measures

16- 'Fixed-effect' or 'Random-effect' models should be written using small letters and not capital letters.

Risk of bias across studies and additional analyses

17- Why did not the authors improve their synthesis of the collected data by doing sensitivity analyses?

18- What about publication bias? Did the author perform any type of 'publication bias assessment'?

Results

19- Why did the author mention that study no 15 (Gibreal et al, 2018) had a discrepancy in the data? What did they mean? Where is the discrepancy? This study has been highly rated in other systematic reviews? I think that they should have contacted the authors before giving such judgement. Also, this study seemed not to be included in any meta-analysis performed in this SR. Actually, the authors are in a great need to synthesize a quantitative measure of the efficacy of piezocision in leveling crowded teeth; they have now several papers to be amalgamated quantitatively.

Risk of bias within studies

20- What is meant by the word 'missingness'. I think that this word should be replaced with a better one.

21- I have many doubts about the accuracy of the given judgments regarding the several domains assessed for each RCT entered into this RISK of BIAS analysis. I really need an explanation for each judgement given in Table II. I am little confused about the methodology employed in giving such assessment. I believe that the authors should provide us with another Supplementary Table showing the rationale for each judgement given to each domain of the five domains of the ROB2 tool (for the whole 13 RCTs). For example, 'Some Concerns' label was given for the 'Measurement of the outcome' in the study conducted by Gibreal et al (2019), although I believe that there is a 'low risk of bias' in this domain for this specific paper. Another example: 'High risk' label was given for the 'Selection of the reported result' in the other study of Gibreal et al (2018) and it is not clear why the reviewers gave this judgement. 'Some Concerns' were also given to the 'Deviations from intended' domain for the studies of Al-Imam et al (2019) and Alfawal et al (2018) with giving any proper explanations for these assessments. We are an a great need to listen the reviewers' explanations for these ratings.

Results of individual studies

22- The subheadings here are not complete. There should be four subheadings: (1) Canine Retraction, (2) En-masse retraction, (3) Incisors' retraction, (4) Decrowding. The authors of this review have enough data to cover these four categories of tooth movement. The data regarding tooth velocity should be mentioned here and not treatment time.

23- another major subheading should appear here 'Treatment Time' and again the reviewers are encouraged to cover the treatment time regarding the four types of orthodontic tooth movement strategies (i.e. canine retraction, incisors' retraction, en-masse retraction, decrowding).

24- I have found that there is some neglect to some important findings regarding lower anterior teeth decrowding reported by Gibreal et al (2018) who found the highest amount of acceleration in the literature (about 59% acceleration). Therefore, the reviewers are asked to shed the light more on these results and to include this study into their meta-analysis on this topic.

25- The inclusion of Wu study in the meta-analysis of the en-masse retraction of upper anterior six teeth is not wise, since this study should have been excluded from the beginning because the sample size is orthognathic patients.

26- The reviewers of this SR are encouraged to elaborate a little bit on the unique study that have evaluated the retraction of upper incisors only (Al-Imam et al, 2019) which was added to this review in its revised version and was published recently (in the last 28 days)!

Patients reported outcome measures

27- Imam should be written Al-Imam et al (the authors are asked to visit the PubMed website to get the correct names of the authors).

28- More information should be extracted from Gibreal et al study (2019) about pain, discomfort, and functional impairments when having piezosurgery compared to the control group. It was nice to do a meta-analysis combining Gibreal's findings with Charavet's findings to see the overall effect on pain.

29- Patients' satisfaction following piezosurgery is not covered although it is a very interesting and important aspect.

Discussion

Summary of Evidence

29- Discussion should include the other types of orthodontic tooth movement strategies, i.e. en-masse retraction, upper incisors' retraction and decrowding movements. The reviewers should not limit their discussion to canine retraction only. Otherwise, this discussion would be very similar to Alfawal et al systematic review in 2016 (which was published in the Progress in Orthodontics journal and is not currently included in this version of the paper).

30- The reviewers should have contacted the authors of the paper no 15 to clarify the problem that they encountered when reading the paper. The reduction of treatment time of decrowding was 59%. Maybe there was a typographical error when the authors of that paper mentioned that the alignment took 3 months in both groups (i.e. the experimental and the control groups). Going back to the tables of that paper would have resolved all the uncertainties.

31- Justification of conducting this SR should be given in the beginning and not in the Discussion section.

32- The discussion regarding patients' reported outcomes has not been updated following the inclusion of recent RCTs. No conclusions were added from Gibreal et al study (2019), nor information about patients' satisfaction was given.

33- The conclusions of the current SR regarding canine retraction velocity are very close to those given by Alfawal et al (2016) in their SR. Therefore, a quick mentioning of that paper is advisable.

Figures Captions

34- The captions are not really explaining everything to the reader. Forrest plots should be mentioned with the names of variables under assessment. Also, the type of tooth movement should be given in the caption (i.e. canine retraction, en-masse retraction, decrowding, etc...).

Figure 01

35- The number of records after removing duplicated was 1728. Then these records were screened. Finally, the reviewers decided to removed 1518 records. Did the reviewers really read and screen 1728 papers? How many difficult hours did they spend in reading the titles of 1728 papers? Unbelievable amount of labor and time being spent in reading the titles of 1728 papers? Is this what happened? I am really confused.

36- Again, after taking out 1518 papers form the retried collection (the 1728 papers), the reviewers should have kept within their hands 210 papers. This not mentioned in the Flow Diagram. How did the number decrease from 210 papers to 30 papers? I need some explanations please.

Figure 02

37- Risk of bias domains. Here, high risk of bias should be given as (-), whereas low risk of bias should be given as (+). Some concerns should be given as (?). This is the general method of presenting Risk of Bias graphs. Please consider reformatting your graph. The same comments are applicable to the second Risk of Bias tool (ROBINS-I), where serious should have (--), moderate should have (-), and low should have (+). No information could be kept (?) or changed to (NA).

Tables: Need to be organized and the authors are asked to use more abbreviations to make their tables more concise (particularly Table 1).

7. PLOS authors have the option to publish the peer review history of their article (what does this mean?). If published, this will include your full peer review and any attached files.

Reviewer #1: No

Reviewer #2: Yes: Assoc. Prof. Nita Viwattanatipa

Reviewer #3: No

Reviewer #5: No

Reviewer #6: No

---

## [Author Response · Author response to Decision Letter 1]

18 Feb 2020

PONE-D-19-31301R1

Is Piezocision effective in accelerating orthodontic tooth movement: a systematic review and meta-analysis

PLOS ONE

Dear Editor Dr. Papageorgiou

Thank you for your kind revision letter. We write this letter to respond to each point raised by the review process.

We have tried our best to address all the points raised by the reviewers. There were some points we couldn’t address from the reviewer 6 and a few of those points also need to your guidance on how to proceed. 

• The reviewer 6 pointed out that in two trials we didn’t applied the risk of bias tool in the right way. We have put forward our understanding how we applied the risk tool. We hope it would satisfy the reviewer. 

• We believe a comparison and explanation between Risk of bias tools ROB2 and ROBINS I is beyond the scope of this article. In our first submission we included some explanation for ROB2 but as we have now included two tools according to reviewers’ suggestions, so we have erased some of our pervious discussion about ROB2 in the methodology so to avoid unnecessary confusion. Also, the suggestion that we should include Kappa statistic data in the review also seems beyond the scope of present study as to the best of our knowledge no systematic view published shared this data. 

• Some of the suggestions of including some studies in introduction and discussion would add nothing in the available information and we do not want to get out into a situation where other reviewers have objections as five of the reviewers are fine with what we have done in this domain.

• We are clueless on answering the reviewer 6 question that how we went through all the articles extracted through the search strategy? 

We are looking forward for your valuable guidance. 

Reviewer #1: The authors have successfully addressed previously raised comments. Authors reviewed the manuscript as suggested.

Authors’ response 

Thank you for your previous comments that improved our manuscript

Reviewer #2: Authors’ efforts in revising their manuscript is appreciated.

MATERIAL AND METHODS

1. Suggestion to rephrase for clear understanding: PICOS

From: Comparison: No adjunct procedures for accelerating tooth movement.

To: Comparison: Piezocision VS Control

2. Please recheck consistency across tables and figures.

Ma’s et al study appears in table II and Fig 2A/ Not appear in Table IV

Because authors’ study focus canine / Anterior retraction, does Ma’s et al study actually fit authors’ criteria?

Should it be included in final full text evaluation?

Ref 30. Ma et al.: orthodontic traction of impacted mandibular third molars

Ma Z, Xu G, Yang C, Xie Q, Shen Y, Zhang S. Efficacy of the technique of

piezoelectric corticotomy for orthodontic traction of impacted mandibular third

molars. Br J Oral Maxillofac Surg. 2015;53(4):326-31. Epub 2015/02/02. doi:

10.1016/j.bjoms.2015.01.002. PubMed PMID: 25638568.

3. Wu et al's study should be noted in discussion about their technique of raising flap.

Ref 39. Wu J, Jiang J-H, Xu L, Liang C, Bai Y, Zou W. A pilot clinical study of Class III

surgical patients facilitated by improved accelerated osteogenic orthodontic

treatments. The Angle Orthodontist. 2015;85(4):616-24. doi: 10.2319/032414-

220.1.

Authors’ response 

Thank you for your comments which improve our manuscript.

1. We have rephrased this 

2. We have excluded this paper accordingly.

3. We have discussed this point in the discussion.

Reviewer #3: Dear authors,

All my questions and comments have not been resolved.

So, first, thanks for doing it.

Abstract (conclusion part)

I don't agree with the "first three months after the surgery".

Please precise according to the type of orthodontic movement.

Please again in the introduction section, it's wrong to say that the surgical incision is performed 1mm to 3 mm deep. According to Dibart et al., who describe for the first time the piezocision procedure, surgical incision must be performed at least of 3 mm. Please correct this.

Materials and Methods

"healthy patients" is not scientific. Please adjust.

Discussion section

Again, please see my previous comments.

The discussion should be mentioned the effect of the piezocision according to the two types of designs: (1) in case of canine retraction and (2) in case of overcrowding treatment.

Authors’ response 

Thank you for your comments which improve our manuscript.

We have made some amendments. But in the literature at least one randomized clinical trial (Uribe) have made 1 mm incisions with piezotome. 

We have restructured the discussion. First three paragraphs are about canine retraction, the fourth is about overcrowding and alignment in piezocision. In rest of the discussion we covered other aspects of piezocision. We tried to made a structured approach according to our primary and secondary outcomes.

Reviewer #5: Additional comment.

The authors are advised to add in the discussion the problems arising from the split-mouth individual studies and the quality of the data deriving from them, which are further used for the meta-analysis. More specifically, these studies suffer from inappropriate statistical methods for the assessment of their raw data.

As long as the original studies did not use and did not follow the criteria as set by the following references, then the results of the meta-analysis should be interpreted accordingly.

a. B Chung, N Pandis, R W Scherer, D Elbourne. CONSORT Extension for Within-Person Randomized Clinical Trials J Dent Res 2020; 99 (2): 121-124.

b. Nikolaos Pandis , Bryan Chung Roberta W Scherer , Diana Elbourne , Douglas G Altman CONSORT 2010 Statement: Extension Checklist for Reporting Within Person Randomised Trials BMJ 2017, 357, j2835.

It is recommended to modify the discussion according to the above comment.

Authors’ response 

Thank you for your comments which improve our manuscript. We would like to thank you to refer us to those articles. We were very glad for this addition to our discussion.

Reviewer #6: Abstract

1- Objectives did not include the 'patient-reported outcomes' as one of the aims of this systematic review. The authors mentioned only 'adverse effects' which is not similar to 'patient-reported outcomes.

2- Search Methods: The English language should be improved in this section.

Introduction

3- The justification of the onset of this systematic review should be mentioned here. The authors are advised to put all of the reasons for conducting this SR at the end of the Introduction section. Here, the authors say claim that no previous systematic review has given the pure effect of piezocision although there is the well-known systematic review of Alfawal et al (2016) published in the Progress in Orthodontics journal which has given some conclusions very similar to the given here in this SR. The authors are also encouraged to include this review in their bibliographic list. Another recent publication (systematic review) assessed different methods of acceleration (mainly piezocision) on the en-masse retraction of upper anterior teeth and this SR should also be visited by the authors to update their introduction, justification and bibliographic list. The paper is written by Khlef, Hajeer, et al (2019) and was published in the JCDP (PMID 31058623).

Objective:

4- The given objective in the text is wrong. The authors should mention that this review aimed to evaluate the available evidence regarding tooth movement acceleration using piezocision (and not only canine retraction). When you have a quick at the retrieved papers, you would immediately discover that the included papers covered a wide range of tooth movements and not only canine retraction.

Materials and Methods

5- Participants: The authors should say 'healthy patients with any Class of malocclusion...' and not 'patients without craniofacial anomalies', since the second phrase is vague and its actual positioning lies in the 'Exclusion Criteria' not the 'Inclusion Criteria'.

We corrected this

6- Intervention. Another mistake appeared here. The 'Intervention' should be 'An acceleration method based on piezosurgery' regardless the orientation of the grooves or fissures. Many piezocision-assisted techniques are based on vertical and horizontal movements of the cutting head. Therefore, this criterion should be corrected.

7- Comparison. The authors should say: 'Patients with malocclusion being treated traditionally without any specific method of acceleration'. The English language should be improved although the authors have claimed that the revised version of their paper has been checked and improved.

8- Outcomes. Again, the authors are a little bit confused. The main outcome should be tooth movement velocity. They have now papers discussing canine retraction, en-masse retraction, four-upper-incisors retraction, and leveling and alignment of lower anterior teeth. Therefore, being confined to only canine retraction is totally wrong.

Exclusion Criteria

9- What is meant by 'Cortication for rapid maxillary expansion'?

10- What is meant by 'Patients taking medications that can affect tooth movement'?

11- According this criterion 'Involving participants undergoing orthognathic surgery', how did the author include the study of Wu et al (2015; PMID: 25347045) in their qualitative and quantitative analysis? According to this criterion, their narrative evaluation should be rephrased again and the related meta-analysis should be omitted. Otherwise, the reviewers are strongly advised to take the results of a recent RCT (under publication) about en-masse retraction written by Khlef et al (2020) that compared flapless corticotomy (by piezocision) with traditional corticotomy (using burs and flap reflection) in terms of tooth velocity and treatment time. Results can be obtained directly from the corresponding author and main supervisor of this research project (Hajeer MY).

12- The authors say 'Non-comparative studies, such as cohort studies'. This is really strange since it is well known that cohort studies can be accompanied with control groups which make them 'comparative studies'. The authors should rephrase this criterion scientifically again.

Information sources

13- How did the author calculate Kappa statistics? Which variables did they include in the Kappa analysis of agreement? Tables are required here as Supplementary Tables.

Data Collection

14- What kind of data types did the author collect according to the prepared data extraction sheets? These should be mentioned here.

Risk of Bias

15- Did the five points appear in the ROB2 tool or ROBINS-I tool? Which points were similar between the two tools? The authors are asked to make the items of each tool clear to the readers.

Summary measures

16- 'Fixed-effect' or 'Random-effect' models should be written using small letters and not capital letters.

Risk of bias across studies and additional analyses

17- Why did not the authors improve their synthesis of the collected data by doing sensitivity analyses?

18- What about publication bias? Did the author perform any type of 'publication bias assessment'?

Results

19- Why did the author mention that study no 15 (Gibreal et al, 2018) had a discrepancy in the data? What did they mean? Where is the discrepancy? This study has been highly rated in other systematic reviews? I think that they should have contacted the authors before giving such judgement. Also, this study seemed not to be included in any meta-analysis performed in this SR. Actually, the authors are in a great need to synthesize a quantitative measure of the efficacy of piezocision in leveling crowded teeth; they have now several papers to be amalgamated quantitatively.

Risk of bias within studies

20- What is meant by the word 'missingness'. I think that this word should be replaced with a better one.

21- I have many doubts about the accuracy of the given judgments regarding the several domains assessed for each RCT entered into this RISK of BIAS analysis. I really need an explanation for each judgement given in Table II. I am little confused about the methodology employed in giving such assessment. I believe that the authors should provide us with another Supplementary Table showing the rationale for each judgement given to each domain of the five domains of the ROB2 tool (for the whole 13 RCTs). For example, 'Some Concerns' label was given for the 'Measurement of the outcome' in the study conducted by Gibreal et al (2019), although I believe that there is a 'low risk of bias' in this domain for this specific paper. Another example: 'High risk' label was given for the 'Selection of the reported result' in the other study of Gibreal et al (2018) and it is not clear why the reviewers gave this judgement. 'Some Concerns' were also given to the 'Deviations from intended' domain for the studies of Al-Imam et al (2019) and Alfawal et al (2018) with giving any proper explanations for these assessments. We are an a great need to listen the reviewers' explanations for these ratings.

Results of individual studies

22- The subheadings here are not complete. There should be four subheadings: (1) Canine Retraction, (2) En-masse retraction, (3) Incisors' retraction, (4) Decrowding. The authors of this review have enough data to cover these four categories of tooth movement. The data regarding tooth velocity should be mentioned here and not treatment time.

23- another major subheading should appear here 'Treatment Time' and again the reviewers are encouraged to cover the treatment time regarding the four types of orthodontic tooth movement strategies (i.e. canine retraction, incisors' retraction, en-masse retraction, decrowding).

24- I have found that there is some neglect to some important findings regarding lower anterior teeth decrowding reported by Gibreal et al (2018) who found the highest amount of acceleration in the literature (about 59% acceleration). Therefore, the reviewers are asked to shed the light more on these results and to include this study into their meta-analysis on this topic.

25- The inclusion of Wu study in the meta-analysis of the en-masse retraction of upper anterior six teeth is not wise, since this study should have been excluded from the beginning because the sample size is orthognathic patients.

26- The reviewers of this SR are encouraged to elaborate a little bit on the unique study that have evaluated the retraction of upper incisors only (Al-Imam et al, 2019) which was added to this review in its revised version and was published recently (in the last 28 days)!

Patients reported outcome measures

27- Imam should be written Al-Imam et al (the authors are asked to visit the PubMed website to get the correct names of the authors).

28- More information should be extracted from Gibreal et al study (2019) about pain, discomfort, and functional impairments when having piezosurgery compared to the control group. It was nice to do a meta-analysis combining Gibreal's findings with Charavet's findings to see the overall effect on pain.

29- Patients' satisfaction following piezosurgery is not covered although it is a very interesting and important aspect.

Discussion

Summary of Evidence

29- Discussion should include the other types of orthodontic tooth movement strategies, i.e. en-masse retraction, upper incisors' retraction and decrowding movements. The reviewers should not limit their discussion to canine retraction only. Otherwise, this discussion would be very similar to Alfawal et al systematic review in 2016 (which was published in the Progress in Orthodontics journal and is not currently included in this version of the paper).

30- The reviewers should have contacted the authors of the paper no 15 to clarify the problem that they encountered when reading the paper. The reduction of treatment time of decrowding was 59%. Maybe there was a typographical error when the authors of that paper mentioned that the alignment took 3 months in both groups (i.e. the experimental and the control groups). Going back to the tables of that paper would have resolved all the uncertainties.

31- Justification of conducting this SR should be given in the beginning and not in the Discussion section.

32- The discussion regarding patients' reported outcomes has not been updated following the inclusion of recent RCTs. No conclusions were added from Gibreal et al study (2019), nor information about patients' satisfaction was given.

33- The conclusions of the current SR regarding canine retraction velocity are very close to those given by Alfawal et al (2016) in their SR. Therefore, a quick mentioning of that paper is advisable.

Figures Captions

34- The captions are not really explaining everything to the reader. Forrest plots should be mentioned with the names of variables under assessment. Also, the type of tooth movement should be given in the caption (i.e. canine retraction, en-masse retraction, decrowding, etc...).

Figure 01

35- The number of records after removing duplicated was 1728. Then these records were screened. Finally, the reviewers decided to removed 1518 records. Did the reviewers really read and screen 1728 papers? How many difficult hours did they spend in reading the titles of 1728 papers? Unbelievable amount of labor and time being spent in reading the titles of 1728 papers? Is this what happened? I am really confused.

36- Again, after taking out 1518 papers form the retried collection (the 1728 papers), the reviewers should have kept within their hands 210 papers. This not mentioned in the Flow Diagram. How did the number decrease from 210 papers to 30 papers? I need some explanations please.

Figure 02

37- Risk of bias domains. Here, high risk of bias should be given as (-), whereas low risk of bias should be given as (+). Some concerns should be given as (?). This is the general method of presenting Risk of Bias graphs. Please consider reformatting your graph. The same comments are applicable to the second Risk of Bias tool (ROBINS-I), where serious should have (--), moderate should have (-), and low should have (+). No information could be kept (?) or changed to (NA).

Tables: Need to be organized and the authors are asked to use more abbreviations to make their tables more concise (particularly Table 1). 

Authors’ response 

Thank you for your comments for improving our manuscript.

1. Yes, we have mentioned it as adverse effects. As piezocision is an invasive procedure so almost all of the patient reported outcomes which have been reported in the literature as we believe comes in adverse category.

2. We improved the language.

3. We gave a justification in the introduction and cited 5 systematic reviews, two of them were published in 2019. We believe adding another SR will add nothing in the flow of information and justification of our SR, also we don’t know if extra citation will be suitable for the journal.

4. Thanks for the input and we have made changes in our objective to make it more descriptive. 

5. We corrected it.

6. We corrected it.

7. We corrected it.

8. We have rephrased the outcomes.

9. We corrected it.

10. We have elaborated this with some examples. 

11. In Wu et al (2015; PMID: 25347045) study rate of tooth movement was measured before orthognathic surgery. So, there was no cofounding factor that can affect the rate of tooth movement. It was an issue if surgery first approach was used. Just to avoid confusion we have made changes in the exclusion criteria. The article Khlef et al (2020) do not comes in our search dates. A previous attempt in April 2019 to get findings of another unpublished study from the same team went in vain. We respect their reservations about not sharing data of an undergoing study and it don’t think it is appropriate for us to contact the same team again for a similar request.

12. We corrected this

13. The two reviewers sent tables for screened papers marked by (Include, Exclude, and Unsure), as stated in Cochrane Handbook, to the statistician and he calculated the Kappa statistics.

14. We collected continuous data 

15. We can easily do this. But in the hindsight, we believe that this is the not scope of this review to dedicate a page or two for the difference between the two tools. As it is now quite some time that the risk of bias tool is being used in systematic reviews so most of the readers know about the new tool. 

But even after this submission the editor or the reviewers felt that we should dedicate some of our writing for the basic difference between the two tools we will gladly include them.

16. Thanks for the input. We have corrected it.

17. We agree with the reviewer that sensitivity analyses improve the certainty in the results, but

there were a few pooled studies in our meta-analysis, so it was not justified. Even that if the editor asked us to include it, we will do that.

18. For the publication bias we do not find it is important to detect three or four trials for publication bias, but we can do that if the editor decided that.

19. This paper was discussed in the discussion section.

For Gibreal’s data.

This is their Outcome Measures “Follow-up of this trial was considered finished when the LII (Little’s index of irregularity) was less than 1 mm, before inserting the first archwire (T0), after 1 month of treatment onset (T1), after 2 months (T2), and at the end of the alignment stage which was accomplished when LII was less than 1 mm (T3)”

In their results: “A statistical significance was found between the two groups regarding the OAT (overall alignment time). The experimental group required less mean treatment time (i.e. 53.5 ± 12.5 days) in the leveling and alignment stage compared to the control group (i.e. 131.4 ± 38.5 days; P < 0.0001) with a 59% decrease in the OAT.”

And this is their table: Little’s index of irregularity (mm)

We noticed that the mean at baseline is similar. After one-month, experimental group showed de-crowding of – 7.45mm and control showed -2.82mm. After the second month experimental group showed de-crowding of -1.37mm and control group showed – 2.49mm. In the third month, So the rest of crowding is 2.75 mm in experimental group and 5.56 in the control group which resolved in both group without defined cut off point. if the cut off point of crowding less than less than 1 mm how the trial concluded that OAT was 53 days. 

Moreover, if the third month should show decrowding similar to the second month such as 1.37 in experimental group and 2.5 in control group. Even if we supposed there is nothing in the data, authors defined T2 as point after two months which mean 60 days that not equal to 53 days as they reported.

For Al imam article, there were three questions to be judge on this domain:

“2.1. Were participants aware of their assigned intervention during the trial?

2.2. Were careers and people delivering the interventions aware of participants' assigned intervention during the trial?

2.3. Were there deviations from the intended intervention that arose because of the experimental context?”

This is the tool Elaboration: “When interest focusses on the effect of assignment to intervention, bias (in this domain) only arises when there are deviations that arose because of the experimental context, i.e. due to expectations of a difference between experimental and comparator intervention.”

The authors wrote “One patient with­ drew from CG for personal reasons and another patient was excluded from the experimental group, because she did not follow the given oral hygiene instructions thoroughly, which caused acute post­surgical inflammation at the palatal side between upper central incisors.’

However, we think the authors made a mistake regarding the cut off point 

“The incisors retraction stage was started by applying coil springs (T0 = start of observation) and considered complete (Tf = end of observation) when one of the 2 possible events occurred: spaces lateral to incisors were closed, or a contact between up­per incisors and lower incisors or the brackets on lower incisors was observed.” 

This cut off points are misleading. we understand that: although there is a remaining space, the treatment is finished because observing lower brackets does not mean contact between upper and lower incisors and does not mean space closure. 

As far as we know that using t test in repeated measures increases the chance of type I error (rejecting null hypothesis when it is true for three time points by probability of 1-(1-0.05)3= 0.14. So, we also have some concerns regarding the statistical analysis in the two articles. So accordingly, we think this domain in in some concern.

20. We corrected it.

21. We corrected it.

22. We corrected it.

23. We corrected it.

24. We have made some concerns on the asked study and we answered this in the point 19.

25. We have modified the exclusion criteria to avoid confusion in this domain.

26. We have included this study as part of our review. We have discussed this paper in detail.

27. Although we get the citation from the authors directly, we have modified that now.

28. We discussed about patient reported outcomes. Within the limit of an article we believe not everything article can be discussed in extreme detail. We will look forward to reviewer and editor guidance on how to further proceed on this aspect. We have no problem in discussing every article in detail.

29. We made a structured approach according to primary and secondary outcomes. We discussed canine retraction, alignment of teeth, en mass retraction, duration of treatment, iatrogenic damages and patient reported out comes like pain etc.

30. Again, this clear in the paper, we have made our point in the previous comments and it is up to the editor to decide.

31. We did that

32. Patients reported outcomes comprises many topics, and we cannot cover all of them in this paper.

33. The authors believe that adding this SR will be extra citation without benefits regarding the scientific content. Also, we have a many SR in the bibliography.

34. We corrected it.

35. Yes, it was a very difficult work and in the methodology of this review we made it clear how we followed things in a systematic and well-established way. 

36. Sorry this was made by a mistake in the figure, but it was correct in the manuscript.

37. There is a new figure developed by Cochrane and we used it.

7. PLOS authors have the option to publish the peer review history of their article (what does this mean?). If published, this will include your full peer review and any attached files.

Do you want your identity to be public for this peer review? For information about this choice, including consent withdrawal, please see our Privacy Policy.

Reviewer #1: No

Reviewer #2: Yes: Assoc. Prof. Nita Viwattanatipa

Reviewer #3: No

Reviewer #5: No

Reviewer #6: No

---

## [Editor Report · Decision Letter 2]

3 Mar 2020

PONE-D-19-31301R2

Is Piezocision effective in accelerating orthodontic tooth movement: a systematic review and meta-analysis

PLOS ONE

Dear Dr Mheissen,

Thank you for submitting your manuscript to PLOS ONE. After careful consideration, we feel that it has merit but does not fully meet PLOS ONE’s publication criteria as it currently stands. Therefore, we invite you to submit a revised version of the manuscript that addresses the points raised during the review process.

I have read the reviewers' comments on your revised submission. Most of them have been dealt with, while others seems to still be open (mostly from reviewer #6). I have selected myself some of the remaining comments and have given some extra feedback on my own about what could be improved. I don't believe it needs to be re-send to reviewers once these have been taken care of, so I'll be overseeing this myself.

We would appreciate receiving your revised manuscript by Apr 17 2020 11:59PM. To enhance the reproducibility of your results, we recommend that if applicable you deposit your laboratory protocols in protocols.io, where a protocol can be assigned its own identifier (DOI) such that it can be cited independently in the future. For instructions see: http://journals.plos.org/plosone/s/submission-guidelines#loc-laboratory-protocols

We look forward to receiving your revised manuscript.

Kind regards,

Spyridon N. Papageorgiou, DDS, Dr Med Dent

Academic Editor

PLOS ONE

Additional Editor Comments (if provided):

I’m not that sure if Figure 4 is very sensible, in terms of methods. You are comparing 1-month and 2-month but for that you are using the same study twice. It would be better to do this purely descriptively or to do it with subgroups (1 subgroup at 1 months and 2nd subgroups at 2 months and then do a between-subsgroups comparison).

Nomenclature: “fixed-effect”, but “random-effects” (plural).

Regarding reviewer #6, comment #17: I agree that you have few studies for sensitivity analysis. You can either have a small comment in the text for not doing any or do some small sensitivity analysis according to (a) inclusion of non-randomized studies, problematic randomization or problematic blinding and see if your conclusions remain the same. Shortly only however.

Regarding reviewer #6, comment #21: The same goes for this comment. If the authors are sure they have made sound judgements about the risk of bias assessment, then additional clarity as the reviewer wishes, would be only beneficial for your paper and for interested readers. You might for example consider adding an extra small table in the appendix that explains the red crossed (high risk domains) of Figure 2A and 2C. .

Regarding reviewer #6, comment #19 & #30: It is for me also not very clear what is the problem, so I don’t get the authors’ justification and am more on the side of the reviewer, who is somewhat confused. T0 are similar, T1 and T2 are clear differences. T3 has no difference, BUT T3 is not a fixed timepoint, so this might be confusing. In any case, the authors should double-check their risk of bias assessment and if they’re sure they can just explain it transparently in the text and it should be no problem.

Regarding the issue about whether all or some patient-reported outcomes are included or not in this review (reviewer #6, comment #29 & #32).

Please be very explicit about which outcomes you are including. “Patients’ experience as assessed by the Visual Analogue Scale VAS” is vague. You can write for example that you include either all patient-reported outcomes or only pain/discomfort. But be clear. And if you leave out any potentially interesting outcomes, then just report this in the discussion, so that readers can see that this outcome was not assessed at all in this review.

I you however include an outcome, you must extract this from all studies and do a meta-analysis if possible (reviewer #6, comment #28)

Please make Figure 2C a little more compact.

Table IIA-B has much information overlap with Figures 2A and 2C. I would suggest keeping one way of presenting these data—maybe theTables.

Table III must likewise be condensed and probably by using more abbreviations. Now it is too long / expanded to be put in a paper.
---

## [Author Response · Author response to Decision Letter 2]

9 Mar 2020

PONE-D-19-31301

Is Piezocision effective in accelerating orthodontic tooth movement: a systematic review and meta-analysis

PLOS ONE

Dear Editor Dr. Papageorgiou

Thank you for your guidance. We write this letter to respond to each point raised in the last revision letter. 

Additional Editor Comments (if provided):

I’m not that sure if Figure 4 is very sensible, in terms of methods. You are comparing 1-month and 2-month but for that you are using the same study twice. It would be better to do this purely descriptively or to do it with subgroups (1 subgroup at 1 months and 2nd subgroups at 2 months and then do a between-subgroups comparison).

Nomenclature: “fixed-effect”, but “random-effects” (plural).

Authors’ Response:

We did it as suggested and made subgroups to assess the acceleration in the first and in the second months. As we aimed in the figure previously to access only the piezocision groups. 

Regarding reviewer #6, comment #17: I agree that you have few studies for sensitivity analysis. You can either have a small comment in the text for not doing any or do some small sensitivity analysis according to (a) inclusion of non-randomized studies, problematic randomization or problematic blinding and see if your conclusions remain the same. Shortly only however.

Authors’ Response:

We have added a small comment in the text for not doing the sensitivity analysis.

Regarding reviewer #6, comment #21: The same goes for this comment. If the authors are sure they have made sound judgements about the risk of bias assessment, then additional clarity as the reviewer wishes, would be only beneficial for your paper and for interested readers. You might for example consider adding an extra small table in the appendix that explains the red crossed (high risk domains) of Figure 2A and 2C.

Authors’ Response:

We have added a supplementary table for the risk of bias justification and hope this will be satisfy.

Regarding reviewer #6, comment #19 & #30: It is for me also not very clear what is the problem, so I don’t get the authors’ justification and am more on the side of the reviewer, who is somewhat confused. T0 are similar, T1 and T2 are clear differences. T3 has no difference, BUT T3 is not a fixed timepoint, so this might be confusing. In any case, the authors should double-check their risk of bias assessment and if they’re sure they can just explain it transparently in the text and it should be no problem.

Authors’ Response:

The reviewer#6 have pointed in his comment#30 to a typographical error done by the authors of that paper mentioned that the alignment took 3 months in both groups? So we do feel he is agreed that the alignment in the paper is 3 months in both groups and this was mentioned in the paper’s table, but in the same time the authors reported in the results that the alignment in the piezocision group was done by 53 days (less than two months). So, we are confused and do feel there is a reporting bias. However, we added this in the discussion transparently and justified the risk of bias in the supplementary table. 

Regarding the issue about whether all or some patient-reported outcomes are included or not in this review (reviewer #6, comment #29 & #32).

Please be very explicit about which outcomes you are including. “Patients’ experience as assessed by the Visual Analogue Scale VAS” is vague. You can write for example that you include either all patient-reported outcomes or only pain/discomfort. But be clear. And if you leave out any potentially interesting outcomes, then just report this in the discussion, so that readers can see that this outcome was not assessed at all in this review.

Authors’ Response:

We made a correction for this point.

I you however include an outcome, you must extract this from all studies and do a meta-analysis if possible (reviewer #6, comment #28)

Authors’ Response:

We already have added the outcome of this study when we included it, but for the pain we reported this outcome descriptively for the four studies which included this outcome.

Please make Figure 2C a little more compact.

Table IIA-B has much information overlap with Figures 2A and 2C. I would suggest keeping one way of presenting these data—maybe the Tables.

Authors’ Response:

We have deleted the figures and kept the tables.

Table III must likewise be condensed and probably by using more abbreviations. Now it is too long / expanded to be put in a paper.

Authors’ Response:

We modified this table using more abbreviations.

---

## [Editor Report · Decision Letter 3]

25 Mar 2020

Is Piezocision effective in accelerating orthodontic tooth movement: a systematic review and meta-analysis

PONE-D-19-31301R3

Dear Dr. Mheissen,

We are pleased to inform you that your manuscript has been judged scientifically suitable for publication and will be formally accepted for publication once it complies with all outstanding technical requirements.

With kind regards,

Spyridon N. Papageorgiou, DDS, Dr Med Dent

Academic Editor

PLOS ONE
---

## [Editor Report · Acceptance letter]

27 Mar 2020

PONE-D-19-31301R3 

Is Piezocision effective in accelerating orthodontic tooth movement: a systematic review and meta-analysis 

Dear Dr. Mheissen:

I am pleased to inform you that your manuscript has been deemed suitable for publication in PLOS ONE. Congratulations! Your manuscript is now with our production department. 

With kind regards,

on behalf of

Dr. Spyridon N. Papageorgiou 

Academic Editor

PLOS ONE